# Complex Sound Discrimination in Zebrafish: Auditory Learning Within a Novel “Go/Go” Decision-Making Paradigm

**DOI:** 10.3390/ani15233452

**Published:** 2025-11-29

**Authors:** Anna Patel, Sai Mattapalli, Jagmeet S. Kanwal

**Affiliations:** 1Department of Neurology, Georgetown University Medical Center, Washington, DC 20057-1460, USA; ajp325@georgetown.edu (A.P.); saimattapalli1@gmail.com (S.M.); 2Department of Biology, Georgetown University, Washington, DC 20057-1460, USA; 3Department of Engineering, Vanderbilt University, Nashville, TN 37203, USA

**Keywords:** associative conditioning, attention, complex sound perception, fish, hearing, learning, memory, neuroethology, social reward

## Abstract

We experimentally demonstrate for the first time via audiovisual associative conditioning, the ability of adult fish to discriminate between complex sounds. We utilize a naturalistic approach to develop and deploy an assay with automated training. We show that freely swimming adult zebrafish can discriminate between constant frequency and frequency-modulated sound types. We analyze behavioral data using established machine learning methods for markerless tracking of animals. We explain zebrafish decision-making behavior based on a published computational model and its neural underpinnings. Our setup can be scaled down or up in size to train small invertebrate species or larger vertebrates, respectively, including mammalian species.

## 1. Introduction

Perception of sounds has played a key role in the adaptation of widely divergent terrestrial and volant species in their respective environments. The aquatic environment as well is rich with biologically relevant auditory signals given the unique characteristics of underwater sound propagation and communication, including the fact that sound travels faster and more efficiently underwater than in air [1,2]. Perception of sounds in the aquatic environment, however, remains understudied. Studies in fish species have largely focused on the structure and function of the peripheral auditory apparatus for transduction of simple sound stimuli, such as pure tone bursts or broadband noise, leaving a significant gap in our understanding of how various fish species perceive and discriminate between complex auditory cues [3,4]. In reality, the aquatic environment presents a rich tapestry of complex sounds that are ecologically significant for many fish species [5,6,7,8]. Fortunately, recent reports on auditory responses to pure and spectrally-complex tones, white noise and courtship sounds, and on the organization of the auditory system are beginning to improve our understanding of sound perception in fishes [2,9,10,11,12].

There are several factors that contribute to the difficulty of studying the underwater soundscape. Sound pressure waves originating underwater are largely reflected at the water–air interface, making it difficult to accurately monitor the underwater acoustic scene from above-water recordings. Hydrophones can be used to record underwater sounds, but intermittent noise and multipath propagation add to the difficulty of localizing sounds and obtaining spectrally clean recordings. Finally, the extent to which sound propagation in water depends on changes in sound pressure vs. particle motion remains a point of contention when establishing sensory mechanisms for its perception [13,14]. Therefore, compared to terrestrial and aerial species, our knowledge of complex sound perception in fish remains limited [15,16,17,18,19,20,21,22,23,24,25,26]. Prey- and predator-generated hydrodynamic sounds and movement cues often include components with frequency modulations and other intricate acoustic features [27,28]. Discriminating between these auditory cues can facilitate quick identification and reaction to threats or opportunities in one’s surroundings and this can be critical for survival in many aquatic species.

One of the few well-studied and compelling examples is that of vocalizations in toadfish (*Opsanus tau*) for social communication, including mate attraction and territorial disputes [29,30,31]. These sounds, often referred to as “boatwhistles” and “grunts,” are characterized by both tonal and pulse-like elements, demonstrating how aquatic species leverage sound complexity to convey critical information. Behavioral studies in *Daniolella*, a small teleost fish whose genus is closely related to zebrafish, demonstrate that individuals of this species can detect and respond to various auditory stimuli, including tones and broadband noise, in the context of social communication. Recent findings also show that *Daniolella* has an ultrafast and very loud sound production mechanism for communication [32,33]. This species uses vocalizations that include high-frequency pulses and low-frequency hums, which are thought to play roles in courtship and agonistic interactions. These studies add to the growing body of evidence that sounds play a crucial role in the behavioral ecology of diverse fish taxa in addition to their well-established importance in the life of aquatic mammalian species, such as whales and dolphins [34,35]. A deeper understanding of complex sound perception in fish can have profound implications for our understanding of the underwater soundscape and how fish species interact and adapt within this domain.

Zebrafish (*Danio rerio*) have emerged as an invaluable model organism for studying sensory perception, including auditory processing. Their small size, rapid development, and transparency during early life stages make zebrafish particularly amenable to the use of advanced neurophysiological techniques, such as two-photon imaging and electrophysiology, but also of behavioral approaches [36,37,38,39,40,41,42]. In addition, the zebrafish’s genetic tractability allows for precise manipulation of neural circuits, providing unique opportunities to link behavior with underlying neural mechanisms [43,44,45,46,47]. Behaviorally, zebrafish exhibit robust responses to visual, touch, chemosensory, and vibratory stimuli, making them well-suited for dissecting the neural and behavioral correlates of sensory perception [48,49,50,51,52,53,54,55]. Details about the natural history of zebrafish, including their environmental and predator–prey interactions have not been fully explored (however, see [56]). Species-specific studies, leveraging the strengths of zebrafish as a model system, can contribute to a broader understanding of auditory perception in the aquatic environment at neural, genetic, and ecological levels.

The natural habitat of zebrafish includes shallow streams and mudflats of the Indian subcontinent [57]. Their habitat is also populated by the highly vocal Indian Bullfrog, *Hoplobatrachus tigerinus*, that feeds on zebrafish larvae [58]. Bullfrog vocalizations can last for nearly one second and are generally repeated five to seven times [59]. They contain both noisy and frequency-modulated acoustic elements. We therefore hypothesized that adult zebrafish perceive and discriminate between complex sounds, such as frequency-modulated (FM) and constant-frequency (CF) sounds. Accordingly, we synthesized FM and CF sound stimuli that were well within the auditory spectrum of zebrafish hearing, as established in previous studies [60]. They were acoustically comparable and contained within the upper and lower bounds of the frequency bandwidth and duration of bullfrog vocalizations. To test our hypothesis, we used both a social reward and an aversive unconditioned stimulus for associative conditioning.

Our earlier studies employed underwater proximity sensors and food reward during operant conditioning [61,62]. These experiments were technically challenging as they employed an electronic prototype in a humid environment. In the interim, one-photon calcium imaging studies demonstrated that larval zebrafish could encode FM sounds at the neural level [10,63]. These studies showed that distinct populations of neurons in the torus semicircularis, the midbrain auditory processing center, exhibit differential activity patterns in response to complex auditory cues. They, together with our earlier work, lend support to the possibility that adult zebrafish can learn to discriminate between complex sounds. As with larvae, adult zebrafish must possess the neural architecture for processing complex auditory signals, though this remains to be thoroughly investigated. The data presented in this study show that adult zebrafish are capable of rapidly learning to discriminate between complex sounds.

## 2. Materials and Methods

### 2.1. Animal Acquisition and Maintenance

Wildtype zebrafish (*Danio rerio*) were bred and reared in a 14:10 light/dark cycle (lights on at 9 AM and off at 11 PM) at 28 °C within the institutional core facility. We used individuals from a stable transgenic line (Huc:GCaMP6) of wildtype zebrafish expressing the calcium indicator GCaMP for potential use in two-photon imaging studies. These lines bred from wild-type strain of zebrafish were obtained from the laboratory of Dr. Harold Burgess at the National Institutes of Health, Bethesda, MD and maintained via established procedures in the zebrafish core facility. All fish used were visually confirmed to have normal appearance and swimming patterns prior to testing. Fish were housed in groups of 5–10 fish per 2.5 L tanks within the Division of Comparative Medicine at Georgetown University Medical Center, Washington, DC. Fish were fed once daily to near-satiation over 5 min with a 1:2 mixture of brine shrimp and dry flake food. Average length and weight of fish were ~2.5 cm and ~0.5 g, respectively. A continuous trickle of filtered habitat water (tap water purified using an Evoqua ROGARD™ Series Polypropylene (Filtration World Inc., St. Louis, MO, USA) 1 µm prefilter followed by two Vantage PTC Carbon filters) was maintained in the tanks. We used a total of 55 fish (21 male; 13 female; 21 undetermined; age range: 3 to 12 months). Thirty-four of these were used for assay refinement (detailed in the Appendix A) and 21 in the finalized sound discrimination assay. A small subset of the 21 (4 males, 3 females) were used for overnight memory consolidation tests.

For experiments, fish were transported to the laboratory in a small tank containing habitat water. Three to four zebrafish were transported from the zebrafish core facility to the laboratory in a separate 2.5 L tank, filled with facility-based habitat water. Care was taken to keep fish in the habitat water during all transport and handling procedures. Individual fish were gently transferred from the maintenance tank, preferably with a small clear Plexiglass plastic scoop, with water into a separate tank for transport and transferred similarly to the test tank to minimize stress resulting from transient hypoxia and handling. Fish were acclimated in the laboratory environment for 15 to 30 min before being gently transferred with a transparent, water-filled scoop to the test tank. Fish were allowed ~5 min for acclimation and tank exploration prior to conditioning. The temperature and quality of lighting in the experimental room generally matched that in the core facility. All procedures were approved by the Institutional Animal Care and Use Committee.

### 2.2. Experimental Setup and Control

Fish were transferred to a clear Plexiglass rectangular test tank (25 × 13 × 16 cm deep) and tested individually. Water level was maintained at 4 to 5 cm to keep fish within the depth of focus of a webcam (Logitech 922) placed 30 cm below the tank. The sides of the tank were padded with foam for sound insulation, and the entire tank was lowered inside a large (40 × 25 cm) wooden box (Figure 1A). Recordings were made under uniform fluorescent room lighting suitable for video tracking; reflections on the screen-side wall were minimized with a white plastic sheet lining the side walls and back-end wall. This also prevented fish from seeing their reflection. An LCD screen (5” Touch screen monitor, ELECROW, Inc., Shenzen, China) facing the transparent front-end (screen-side) of the test tank was positioned so that it was pressed against the tank wall within a foam holder. The screen served as a virtual window between the outside of the tank and fish swimming freely in the tank. Both desirable and aversive video clips were displayed on the same LCD screen following sound stimuli as described below. A Logitech (Lausanne, Switzerland) webcam (model 920, HD 1080p, capable of recording at 30 frames per second, fps) continuously recorded fish movements at 10 fps given the low-light recording condition. A small Bluetooth speaker (EWA, model A106 pro; Guangdong, China) was placed 37.1 cm underneath the center of the tank and used to present sound stimuli. Fluorescent room lights provided uniform illumination within the tank for off-line tracking of fish movements. The tank location in the room was adjusted to minimize reflections on the screen-side of the tank. For testing auditory discrimination ability of zebrafish in our setup, we exploited the natural behavior of freely behaving animals. We reasoned that in their natural environment animals can move towards sources of stimuli that are attractive and away from those that are threatening. Therefore, in contrast to the Go/No-Go training paradigm where animal movement is inhibited [64,65], we let zebrafish swim freely when encountering either a rewarding or fearful stimulus (Figure 1B).

Complex sound discrimination was tested using two novel, synthetic sounds. One sound type was a multiharmonic constant frequency (CF) tone burst embedded within a background of narrowband noise and the other was a multiharmonic frequency-modulated (FM) sweep (Figure 2). The amplitude of both types of sound pulses was matched using the root mean square method. The complete noise-embedded CF (NCF) sound type consisted of five 1 s duration tonal pulses, with a fundamental of 500 Hz and a bandwidth 500 to 1500 Hz across three harmonics, punctuated by a brief silence interval between each pulse so that the total sound duration was 6 s (Figure 2A). The frequency in the FM sound was rapidly modulated downwards (at a rate of 5 Hz per ms) with three harmonics within a bandwidth ranging from 50 to 1500 Hz (Figure 2B). This complete FM sound type consisted of six, 500 ms downward FM (DFM) sweeps with a total sound duration of 6 s. In summary, stimulus selection (NCF vs. DFM) was constrained to zebrafish hearing range and matched in energy; FM rates were designed to be consistent with amphibian vocalization envelopes, providing ecologically equivalent complexity while remaining synthetic to avoid familiarity confounds.

Figure 2C,D show the spectrograms of both sounds as recorded with a hydrophone (DE-PRO, DolphinEar Global) with an omnidirectional configuration and a wide frequency-response of 1 Hz to 24 kHz. The spectrograms were plotted using Raven Pro, vers. 1.6 (Cornell Labs, Ithaca, NY, USA) software. Details of the procedure and frequency-response curves recorded under water for the Bluetooth speaker are provided in the Appendix A. The underwater sounds were attenuated by ~14 dB in amplitude. Sound transmission through the Plexiglass and water also introduced a low-level of low-frequency noise that was picked-up by the hydrophone in both sounds. The second harmonic in the NCF was relatively pronounced and the noisy component was reduced in amplitude but otherwise did not show any significant distortions in the frequency structure.

### 2.3. Experimental Paradigm and Control

Our behavioral paradigm uses a “Go-to/Go-away” associative learning framework in which the fish learns to differentially respond to two auditory stimuli by either approaching or withdrawing from a visual target screen. Together with markerless-tracking, this design enables measurement of continuous behavioral trajectories rather than binary choices, allowing one to characterize the moment-to-moment evolution of behavioral state dynamics during learning. Rather than forcing a correct/incorrect response at a discrete time point, the paradigm preserves the fish’s natural ability to swim freely, producing rich, time-resolved behavioral signals that allow quantification of learning in terms of trajectory shape, spatial proximity, and steering dynamics. For associative conditioning within the Go-to/Go-away paradigm, an auditory stimulus was followed by either a rewarding or an aversive visual stimulus after a ~3 s silence or “gap” interval. The NCF sound was paired with a socially desirable visual stimulus: a video of a cohort of freely swimming zebrafish displayed on the LCD screen. The DFM auditory cue was paired with an aversive visual stimulus—a video of an Indian Bullfrog, *Hoplobatrachus tigerinus*, with inflating/deflating vocal sacs displayed on the same screen. The Indian Bullfrog is a voracious, generalist predator whose diet includes small vertebrates and sometimes fish [58]. It inhabits the same habitat as zebrafish although there is no direct evidence of their predation on zebrafish. Screen luminance and placement were constant across US conditions and analysis windows exclude the US epoch. Both videos were displayed below the water level in the tank for fish to have a clear visual. For automated playback, both sounds and videoclips were read from files saved on the computer’s hard drive.

A key feature of the “Go-to/Go-away”, abbreviated as the “Go/Go”, training paradigm is that it involves attraction to and repulsion from a stimulus presented at the same location, i.e., the response exhibited to one CS-US pair is opposite to that of the other CS-US pair. Within this paradigm, fish swim either towards or away from the LCD screen (depending on the CS-US combination presented) during the gap interval between the CS and US. Details of the development of this methodology and intermediate results are described in the Appendix A. In brief, assay development started with a single sound stimulus block consisting of eight repetitions of the same CS-US pair. In separate animals, either the N (CF-aversion and DFM-reward or NCF-reward and DFM-aversion pairing was used. Exploratory free-swimming patterns were tracked during initial and final baseline conditions to be able to gauge the effect of conditioning (see Appendix A). Success with single CS-US pairing estimated from density plots for distance from screen (described in Appendix A and illustrated in Appendix A) was followed by a two-block design consisting of eight repetitions of NCF-reward (Block 1) and DFM-aversion pairs (Block 2) within a single conditioning session in the same animal. Here, we first consecutively presented the same CS-US pair six times (Block 1) before switching to six repetitions of the alternate CS-US pair in Block 2. Thus, each training session included six consecutive presentations of either the CF or FM audio/visual pair (Block 1) followed by another six consecutive presentations of the alternate pair (Block 2). The order of presentation of the CF or FM audio/visual pairs was randomly selected across fish. These experiments were used to study block interaction effects on conditioning efficacy (see Appendix A).

In the finalized design, a third block consisting of three NCF-reward (abbreviated as NCF*_rew_*) were randomly alternated with three DFM-aversion/fear (abbreviated as DFM*_fear_*) runs. Tracking data from these last six runs (#13 to 18) were used to test sound discrimination in 21 adult zebrafish. Thus, each animal was exposed to a total of eighteen runs within a three-block (six runs per block) design per session and tested only once. The duration and timing of presentation of the CS and US were the same in all runs (see Figure 3). Our primary readout is behavior during the 3 s gap after the CS and before any video appears; thus, the discrimination metric is independent of US brightness. Within-subject counterbalancing (block order randomized across fish) further controlled for any residual visual bias, such as subtle differences possibly in US brightness. In a subset of fish (n = 7), a second contracted (single block) session was used the next day to test for overnight memory consolidation of response to the NCF and DFM sound presentation across different individuals.

All experiments were conducted under computer control using a desktop computer (Dell PC running Windows 11 software). We used Python (vers. 3.10)-based custom-written code (using Win32COM, OpenCV and Tkinter) to control all aspects of the experiment. A programmable microcontroller delivered the aversive visual stimulus displayed on an LCD screen on one side of a test tank. In each experimental run, a 10 s baseline was followed by the auditory stimulus, which was played for 6 s. After a brief (~3 s) delay, the sound was followed by the corresponding rewarding or aversive stimulus. A 3 s delay was chosen based on previous observations on the time it takes for fish to respond to a conditioned stimulus [61]. The trial was repeated after a randomized inter-trial interval (ITI), which ranged from 100 to 200 s. Before and after each session, there was a 6 min period in which the free-swim patterns, without presentation of any stimulus, of fish were video-recorded and labeled as baseline activity. Each single-day session lasted approximately 2 h, after which fish were returned to the home tank. For two-day sessions, Block 1 runs of retraining, consisting either of NCF*_rew_* or DFM*_fear_*, were administered once again the next day.

### 2.4. Animal Tracking

Machine learning software ZebraZoom (vers.1.6) [66] was used to track free-swimming zebrafish behavior, tracking the head and tail coordinates of the moving fish per frame. The tracking software was trained using configuration files of 3–4 frames of manually identified fish head and tail tip tracking points. Optimization options within ZebraZoom were used to ensure the coordinates of the zebrafish were recorded throughout the entire session consisting of multiple trials or “runs”. These algorithms can be trained to track one or more fish across thousands of frames of a video. They provide a new way to quantify behavior, continuously tracking head orientation and fish locations. We extracted an Excel spreadsheet with per-frame coordinates for both head and tail markers in the x-y plane, as well as values for heading direction.

**Figure 3 animals-15-03452-f003:**
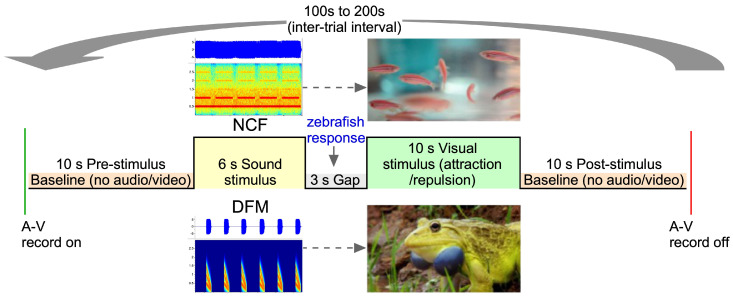
Timeline of events during a conditioning trial. Video clips of swimming zebrafish (desirable/rewarding US) and Indian Bullfrog (aversive or fearful US) were displayed on an LCD screen after a 3 s gap interval following the end of a 6 s long complex sound (NCF or DFM). A-V: audio-video; NCF: noisy CF sound; DFM: downward FM sound. Frog video clip was adapted from https://share.google/I6GxVJZEvhvlJfaJt, accessed on 1 October 2022.

### 2.5. Data Analysis and Statistical Methods

Distance of a zebrafish from the screen after sound presentation was estimated from continuously tracking x-y coordinates of fish locations using machine learning algorithms, such as ZebraZoom that also computes heading. From these data, one can compute velocity, acceleration, and related swim parameters. An animal’s ability to predict the arrival of the visual display (unconditioned stimulus or US) from the sound (conditioned stimulus or CS) was taken as evidence of perception and discrimination between the sounds. Unlike the Pavlovian-conditioned autonomic reflexes, however, we observed the naturalistic behavioral response of either swimming towards or away from the placement of the US. Thus, the average distance from the screen during the gap between the end of the 6 s long NCF or DFM sound and before the video playback on the LCD screen served as an absolute measure of attraction towards or repulsion away from the US. A directional difference in successive positions of each animal could also serve this purpose.

Statistically, the experiment was structured as a within-subject repeated-measures design. Each fish experienced both stimuli across matched trial blocks, permitting paired comparison of behavioral state evolution rather than between-group contrasts. This approach increased statistical sensitivity and reduced inter-subject variability. Thus, the Go-to/Go-away paradigm provided the continuous kinematic structure necessary to evaluate the stability and learning-dependent reorganization of behavioral policy. For the final discrimination test performed with n = 21 fish, the sample size was determined based on prior behavioral work in adult zebrafish examining conditioned approach/avoidance responses [61,67] and deemed typical for within-subject behavioral learning paradigms in adult zebrafish.

Overnight consolidation was quantified as changes in state-space dispersion in Block 1 on each day. The small subset (n = 7) tested for memory consolidation consisted of fish drawn from the same trained cohort. This constituted a retention confirmation assay rather than an independent inferential test, consistent with prior zebrafish-conditioning studies (e.g., [68]). We quantified state-space dispersion within the first six trials of each day (Block 1), which representing nascent behavioral state. For each fish, we computed the variance of proximity (distance to the goal) and steering (yaw), aligned to the direction of the conditioned response, and used the trace of the covariance matrix as a scalar measure of the total variance in exploratory or conditioned behavior. A high trace indicates a high total spread across all variables, while a low trace indicates a low spread.

Statistical analyses were largely performed using JMPpro (vers. 8.1, SAS Inc.) and custom Python scripts (vers. 3.1). To eliminate edge effects when analyzing positional information, we ignored fish locations 1 cm from the backend and 2.0 cm from the screen-side of the tank to minimize bias from fish lingering on screen-side because of possible screen reflections and increased engagement due to the backlit display. One fish that behaved in an anomalous manner, probably due to anxiety effects, was eliminated from the two-day testing data. Density and box plots were used to visualize high density fish locations across the two stimulus conditions. We performed multivariate discriminant analysis (DA) to test for significant behavioral differences (Wilks’ lambda) between the response to the presentation of CF and FM sound types. We visualized single parameter differences as boxplots and deployed nonparametric (Wilcoxon) tests of positional information. Effect size was calculated using Cohen’s *d* for two samples and significance level was set at 0.05 for all analyses. GenAI was used during initial literature search, to assist with parts of data analysis and interpretation and to refine text at a few locations.

## 3. Results

### 3.1. Learning Dynamics During Associative Conditioning

We used distance from screen as a measure to compare fish locations for the two sound cues. On average, fish moved towards the screen-side of the tank during the gap interval after the NCF sound cue and away from the screen in response to the DFM sound cue. Fish location data were first obtained for eight runs each during dual stimulus conditioning and plotted as density plots for each sound type (Appendix A). NCF*_rew_*- and DFM*_fear_*-conditioning sessions, shown as density plots, illustrate the “centers-of-gravity” of fish locations (Figure 4A). The density plots show that even within the 3 s interval, fish are more likely to be present near the screen for the NCF*_rew_* sound cue and away from it for the DFM*_fear_* sound. This suggests, on average, a high level of activity of fish moving towards or away from the screen within a short timeframe. Over successive runs, fish gradually learn to predict the US that will be presented on the screen in response to each sound cue (Figure 4B). To track run-to-run variability, we also examined fish distance from screen for both sound types. As seen in the spline plots, overall fish behavior during learning did not always follow a linear trend. This is likely due to waxing and waning of attention and possible hysteresis effects. Further details are included in Appendix A. Divergence in swim direction begins to appear by the third run and is maximal by the tenth training run. Overall, fish tend to show some adaptation to the DFM*_fear_* stimulus in the eleventh and/or twelfth run.

### 3.2. Post-Conditioning Swim Trajectories

We conducted single-fish trajectory analysis after the first twelve conditioning runs, consisting of a block of six NCF*_rew_* runs that were either preceded (in one group) or followed (in the other group) by a second block of six DFM*_fear_* runs. In the thirteenth run, their behavior was tracked as they traversed the tank space during the 3 s gap interval immediately after presentation of either the NCF or DFM sound type. Trajectories of those conditioned for NCF*_rew_* in second block were plotted separately (Figure 5A) from those conditioned for DFM*_fear_* (Figure 5B). Sample responses are shown as color-coded tracks in six animals for each sound type. Fish positions were tracked offline using ZebraZoom (vers. 1.6) from recorded videos. Each track spans ~3 s after the sound ends.

In response to the presentation of the NCF*_rew_* sound, fish tended to swim towards or stay near the screen, e.g., ZF160, ZF166, and ZF167 (Figure 5A). ZF172 moves away at first, which can happen if the previous sound cue is fearful, but pauses and turns to swim towards the screen. In response to the presentation of the DFM*_fear_* sound, fish generally stay away or swim away from the screen (Figure 5B). Three of the six fish shown, however, stay at about the same distance from the screen and one (ZF161) even moves towards the screen. While the behavior of some fish adds noise to the data, overall fish remain further away from the screen in response to DFM*_fear_* sound compared to their positioning in response to the NCF*_rew_* sound.

### 3.3. Behavioral Evedence for Complex Sound Discrimination

We first generated the proportion of density plots separately for the conditioning (Block 2) and test runs (Block 3) to estimate the relative density of fish locations for NCF*_rew_* vs. DFM*_fear_* sound cues during the gap interval (Figure 6A). This exploratory analysis provided us with a region of interest within the tank length where the fish appeared to be differentially active in response to the two sound types.

The proportion of densities exhibited a steep gradient between 50 and 150 mm from the screen where the relative NCF/DFM response densities of fish locations changed rapidly (Figure 6A). This pattern was similar for the second and for the third block (4848 frames each across six test runs per fish). We labeled this range as a “decision-making zone” where fish most frequently transitioned either towards or away from the screen following each sound presentation per the CS-US-conditioned prediction of the US. We, therefore, restricted analysis of behavioral response parameters to this zone of the tank. A comparison of Block 2 boundary (solid line) between NCF/DFM densities vs. Block 3 runs (dashed line) of the relative density of fish positions yielded a difference (hatched area) of 150 and 200 mm (in Figure 6A). We labeled this bump as a “region of behavioral uncertainty” where fish apparently moved slightly away from the screen in response to the NCF*_rew_* sound, possibly because of the lingering or hysteresis effect of a previous DFM*_fear_* sound presentation as explained in greater detail in the Appendix A.

For a robust estimate of learning dynamics, we compared the distribution of fish locations during Block 2 runs of CS-US conditioning for NCF*_rew_* and DFM*_fear_* pairs during the gap vs. the pre-stimulus interval. We selected Block 2 runs for this comparison because here each block contained the results of six consecutive presentations of the same sound cue in contrast to randomly alternating ones in Block 3. Both NCF*_rew_* and DFM*_fear_* runs showed a clear difference in fish locations compared to the pre-stimulus presentation condition (NCF*_rew_* = 1146 frames and pre-stim = 2743 frames: DFM*_fear_* = 1755 frames and pre-stim = 4336 frames). Median values for pre-stimulus vs. post-NCF*_rew_* presentation conditions were 88 and 83.7, respectively, and the pre-stimulus vs. DFM*_fea_* presentation conditions were 94.1 and 97.6, respectively.

For testing a difference in fish behavior in response to the two sounds and hence their ability to discriminate between them, we performed multivariate discriminant analysis (DA) by pooling three behavioral response parameters for test runs (13 to 18). Directional bias was computed by multiplying the magnitude of change in fish displacement (positive towards the screen and negative away from the screen) with the swim-speed computed from the change in the x-y coordinates of the head location. For swim activity > 16 mm/s, successive video frames showing a lower swim rate were discarded to estimate and compare active swim locations during the gap interval. These data yielded a significant effect of sound type (Wilks’ Lambda = 0.942, F (3, 216), *p* = 0.0048) (Figure 6C). ROC curves gave an area under the curve of 0.634 for both 103 CF and 118 FM presentations. There was a 41% misclassification with a −2 log likelihood score = 363.33. The group means for each parameter are shown in Table 1, whereby additional parameters, such as velocity and acceleration, did not substantially improve model performance.

Next, we examined the independent contribution of each response parameter included in the DA to test for significant differences for the two sound presentations. Tracking fish locations across 5712 frames (CF = 2822 and FM = 2890 frames) yielded a difference of 1.43 cm between mean locations in the response to the two sound types presented though the effect size was small (0.20). These data are illustrated as box plots for the 50 to 150 mm range in Figure 6D. Comparing paired CF and FM frame-matched responses (runs 13–18) across all fish yielded 510 paired observations per variable (within six matched CF and FM runs per fish). Zebrafish showed significantly reduced distance from the screen during CF runs relative to FM runs (Wilcoxon signed-rank test, W = 58,179.5, two-sided *p* = 0.043; rank-biserial effect size (r_rb_) = −0.10). Interestingly, there was no significant difference between mean distance from screen values for the 6 s time interval during which the two sounds were presented (see box plots in Appendix A). This suggests that fish do not initiate movement towards or away from the screen reflexively (e.g., startle response) or spontaneously. Instead, they attend to all repetitions within the sequence of sounds before responding, i.e., they wait for the sound to end, allowing for perception, recognition, and prediction of the US during the silence (gap) interval.

Directional bias was also greater during CF trials, consistent with approach versus withdrawal, with a significant directional effect (one-sided test, *p* = 0.025; two-sided W = 54,009.0, *p* = 0.050; (r_rb_) = –0.10). Across pooled frame data, directional bias showed an effect size = 0.47 for the six test runs (Block 3) with a group mean CF = 6.723 and FM = −15.244. (Figure 6E). Not surprisingly, this improved when comparing the last three runs from Block 2 that included a more consistent stimulus presentation (either CF or FM in runs 10 to 12), making it easier for the fish to predict the US. For the level of “swim activity > 16 mm/s”, randomized test runs alone yielded a slight difference between medians and response distributions to the two sound types. Median displacement values after pooling all frames yielded a group mean CF = 21.891 mm/s across 157 frames and FM = 15.315 mm/s across 163 frames (Figure 6F). Effect size was = 0.46 with variances between the two groups being significantly different (*p* < 0.0001, Levene test). When the analysis was repeated across fish using per-fish median effects, second-level Wilcoxon tests were not significant for this (Wilcoxon W = 200.0, *p* = 0.516) and other parameters (*p* > 0.50), reflecting substantial inter-individual variability in effect magnitude for CF and FM presentations. Together, these data indicate that zebrafish exhibit cue-specific approach vs. avoidance behavior, but that this response is variable across individuals, during an early-stage learning timeframe.

As a final test, we computed the mean trajectory direction (encoded as positive towards the screen and negative away from the screen) and duration for the longest trajectories for each animal during each gap interval. We compared the mean of the longest trajectories in response to the CF and FM sounds across all animals (Figure 7A). CF responses were positive, and FM responses showed a negative mean value that was significantly different (*p* = 0.0062, n = 21; Wilcoxon two-sample rank sums test). Effect size was = 0.67 and variances between the two groups were not significantly different (*p* = 0.0722, Levene test). This estimate of directed swimming provided a behaviorally meaningful comparison that did not depend on static fish locations and conformed to general swim patterns observed during the gap interval (see Figure 5). Figure 7B,C shows frames taken from the recorded video clips (see Appendix A) as examples of directional swimming in response to CF and FM sounds. Arrows show the direction and range of swimming in each case. In some cases, such as CF sound example, the fish starts to swim towards the screen at the end of the sound stream and turns back before re-orienting and swimming back towards the screen during the gap interval. This scenario captures a potential source of noise in the data based on where a fish happens to be located at sound onset and offset. If due to early recognition of the CS, the test animal is already moving towards or away from the screen, then a perceived delay in the expected arrival of the US can alter its learned trajectory. Together, these data indicate that zebrafish exhibit cue-specific approach vs. avoidance behavior, and this response is variable across individuals.

### 3.4. Overnight Memory Consolidation

To test the long-term effect of associative conditioning learning, we tested seven fish within same conditioning paradigm the following day to see if there was any effect of overnight memory consolidation. The procedure during the single-block testing session for overnight memory consolidation was the same as the single-block (Block 1 or 2) training sessions conducted the previous day. When comparing data across the first six runs of conditioning, the fish tend to have an advantage on average by starting closer to the screen for the CF sound and further away from it for the FM sound. This is shown by the slope of the best fit regression line for each conditioning paradigm for the Block 1 set of six runs (Figure 8A,B). Effect sizes for change in distance from screen for both types of conditioning were low (0.18 for NCF*_rew_* and 0.3 for DFM*_fear_*), given the small number of animals on which these tests were conducted.

For a final consideration, we compared the final conditioned performance (#s 13 to 18) on the first day with the retention during initial runs (#s one to six) on the following day (Figure 8C). These data obtained after pooling fish locations in all frames, shown as box plots superimposed on smoothened density plots, show that on average, fish retained their memory for sound discrimination based on their difference between mean distances (*p* < 0.01, n = 7). The difference in response to the NCF vs. the DFM sounds relative to the test runs of the previous day was less pronounced (difference in median distances from the screen during the gap interval on day 1 was 11.0 mm and on day 2 = 3.5 mm), suggestive of some level of forgetting.

For an across-animal statistical test for memory consolidation, we used a trial-block state analysis approach where each fish contributed a high-dimensional behavioral trajectory. We observed a consistent reduction in dispersion between Day 1 and Day 2 (Wilcoxon signed-rank, *p* = 0.25; rank-biserial effect size (r_rb_) = 0.80; n = 4), reflecting a large effect size and a strong and consistent directional shift in behavioral variability across fish on day 2 (see Appendix A). This statistical result reflects strong within-individual consistency. In such within-subject designs, statistical power is governed by consistency across individuals and not sample size in the classical between-group sense. This analysis indicated that behavior became more stable and less exploratory after the overnight period, even when mean approach distances remained comparable. Such a reduction in trial-to-trial variability is a well-established signature of memory consolidation and policy stabilization in both motor and associative learning frameworks (e.g., [69]). Here, learning is inferred from shape and stability of behavioral manifolds rather than binary performance metrics.

## 4. Discussion

### 4.1. Sound in the Aquatic Environment

A relatively large body of work underscores the ecological and evolutionary significance of acoustic communication in aquatic environments for mammalian species, but less so for teleostean species. The aquatic environment is acoustically different from terrestrial and aerial environments in that sound propagates nearly five times faster in water than in air and with less attenuation over distance. These properties create a dynamic underwater soundscape rich in biologically relevant signals, including abiotic sounds from flowing water and biotic sounds from conspecifics, predators, and prey. In tropical freshwater ecosystems, such as those inhabited by zebrafish, acoustic signals play a pivotal role in survival and ecological interactions. Fish must detect and discriminate between complex sound patterns for communication, navigation, and predator avoidance [57,70]. For instance, prey-generated hydrodynamic sounds and predator vocalizations often include noise bands, FM components, and other acoustic features that require specialized auditory processing mechanisms [71]. Their perception and processing have not been studied in much detail beyond the peripheral hearing apparatus.

Here we studied complex sound discrimination in zebrafish since this species lives in shallow, acoustically active freshwater habitats of the Indian subcontinent, where they coexist with vocal predators like the Indian Bullfrog (*Hoplobatrachus tigerinus*) [58]. This acoustically rich environment would have led to the creation of neural circuits capable of detecting and interpreting complex auditory stimuli. The present study took advantage of sound stimuli that partially resemble ecologically relevant signals to facilitate conditioning. Our findings underscore the importance of sound discrimination for predator–prey interactions and social behavior in zebrafish, as also observed in related teleosts, e.g., toadfish and *Daniolella* species [33,72,73]. We provide compelling evidence that, similar to mammals, zebrafish can perceive and discriminate between complex sounds, specifically distinguishing an NCF tonal sequence from a DFM pulse sequence, and integrate auditory cues within decision-making processes [74,75].

Unlike land vertebrates, zebrafish and most other fish species rely on a combination of auditory and lateral-line systems for perception of environmental sounds [76,77]. The lateral line is known to complement auditory perception by detecting low-frequency hydrodynamic cues, which are critical for locating predators or prey and for shoaling behavior [78,79]. The demonstration of overnight memory consolidation in zebrafish highlights the robustness of their associative learning and memory systems, positioning them as an ideal model for studying auditory perception and its neural underpinnings.

### 4.2. Hearing in Fishes

Studies of fish hearing have revealed a remarkable range of auditory capabilities, from the specialized sensitivity in some fish species to the broader tuning in others, such as zebrafish. Unlike terrestrial vertebrates where hearing results from changes in air pressure, fish-hearing organs are also sensitive to particle motion within the aquatic medium. One taxanomic group, otophysan fishes (e.g., goldfish, *Carassius auratus*), and clupeids (e.g., herrings), have well-developed hearing that is facilitated by mechanical connections between the swim bladder and the inner ear [1,80]. Among otophysan fishes, goldfish have been a model system for investigating auditory sensitivity and frequency discrimination. Goldfish exhibit high auditory sensitivity across a wide range of frequencies (100 Hz–4 kHz), facilitated by Weberian ossicles, which transmit swim bladder vibrations to the inner ear [1]. Similarly, the mormyrid elephant fish (*Gnathonemus petersii*), another specialized auditory species, demonstrates advanced temporal processing abilities, allowing it to detect fine temporal features of sound, which are critical for electrolocation and communication [81]. Fishes also employ unique auditory mechanisms in specific ecological contexts. For instance, toadfish (*Opsanus tau*) produce and perceive complex vocalizations, including boatwhistles and grunts, which are used in territorial displays and mate attraction [82]. These vocalizations emphasize low-frequency components, well-matched to the toadfish’s auditory sensitivity. Similarly, members of the genus *Daniolella*, closely related to zebrafish, communicate using broadband vocalizations, including high-frequency pulses and low-frequency hums, which likely play roles in courtship and aggression [83]. These studies underscore the diversity of hearing capabilities among fishes and their importance in various ecological and social contexts. Most importantly, they document the need for fish species to learn to recognize and discriminate between complex sound in their environment.

In contrast to auditory specialists, many teleost species, including zebrafish (*Danio rerio*), exhibit broader auditory tuning and, as we show, are capable of complex sound perception and discrimination. Zebrafish have an auditory range between 100 Hz and 2.5 kHz, with sensitivity to low-frequency sounds associated with conspecific communication and predator detection [3,4,60]. Behavioral studies in larval zebrafish have demonstrated their ability to respond to auditory cues in the context of predator avoidance and social interactions [84], but in adults have focused only on behavioral conditioning for discriminating pure tones [3].

In addition to the use of behavioral methodologies, advances in imaging techniques have recently contributed to understanding auditory processing and its evolutionary origins. For example, two-photon calcium imaging in zebrafish has allowed researchers to map auditory responses at the single-neuron level, elucidating how sound features are encoded in the brain [85]. Electrophysiological recordings in goldfish and other species have similarly revealed how auditory neurons process temporal and spectral sound features [76,80]. These approaches have significantly advanced our knowledge of the neural mechanisms underlying auditory perception in fishes and support our behavioral observations.

### 4.3. On-the-Go Learning

Associative conditioning in rodents and zebrafish can take thousands of trials to reach a >75% success rate. In our setup, zebrafish exhibit rapid learning with only five to six runs. This rapid acquisition cannot be explained by a single binary decision enforced across trials, as in many traditional assays [86]. In a safe laboratory environment, mistakes do not have serious consequences but degrees of freedom for movement are severely restricted. Our paradigm allowed animals to swim freely under relatively unconstrained conditions, where behavioral noise and directional errors are common. For example, zebrafish have an intrinsic tendency to randomly circle back-and-forth within their environment. This may reflect exploratory tendencies, predator-avoidance strategies, or periodic verification of predictions. Similar “anxiety-like” edge-tracking behaviors have been described in other fish species [87]. These seemingly erratic swim patterns are largely natural but complicate detection of learning, making the adoption of machine learning (ML) and markerless tracking approaches essential [66,88]. Quantifying behavior within a spatial and behavioral time window allowed us to track subtle changes in behavioral patterns and obtain reliable evidence of learning.

Zebrafish are highly social, exhibiting both schooling and shoaling behaviors, and preferentially approach a shoal of swimming zebrafish, animated images, and even realistic 3D-printed models of conspecifics [89,90,91,92]. In fact, zebrafish become stressed in isolated environments and visualization of conspecifics can alleviate the stress of isolation [84,93]. The Go/Go assay allowed animals to respond freely while minimizing handling stress. Testing of multiple animals simultaneously, while ecologically relevant, can introduce noise due to inter-individual interactions and errors from markerless tracking [94].

### 4.4. A State-Based Model for Auditory Learning

To understand the source of noise within Go/Go paradigm-driven zebrafish behavior at the neural level, we consider a state-based model of auditory learning [95]. The dynamic logistic regression model accounts for trial–trial variability [96]. This variability can result from periodic shifts in covariates, such as attention, motivation, and prior experience that can affect the pattern of swimming (Figure 9A). In this framework, apparent noise reflects transient shifts in internal states that shape the probability of a response to a stimulus. Early in training, zebrafish responses may appear reflexive or exploratory, but these state transitions are critical for establishing long-term associations. Without them, consolidation of auditory–visual learning could be impaired. Switching among latent internal states, as in switching state-space models, provides a useful formalism for capturing time-varying internal dynamics [97,98]. Early shifts in internal states reflect learning effectiveness and the ability to quantify them can be used to assess the effectiveness of interventions, such as drug treatments targeting memory or neural plasticity.

As training progresses, behavioral responses become more deliberate, reflecting adjustments in synaptic weights across sensory–motor circuits. The error corrections in behavioral responses facilitate approach and avoidance behaviors (Figure 9B). The trial-to-trial adjustments in synaptic efficacy, consistent with dynamic models of neural plasticity, play a central role in the consolidation of memory. Our data indicated a qualitative shift from a more widely distributed and variable state-space occupation on the first day of training to a more compressed and directionally focused state on the second day. This reflects reduced exploratory variance and a movement toward a more stable and efficient approach policy. When responses were collapsed to per-fish median values, across-fish Wilcoxon tests were not significant (*p* > 0.25), indicating that individual-level variability is high when temporal sequence information is removed, consistent with an early learning stage in which cue–response associations are emerging but not yet stabilized. Even though paired sample medians did not differ significantly across days, a reduction in state-space dispersion is consistent with partial consolidation of the learned mapping between auditory cue and directional motor response. This is important because consolidation often reduces variance before changing means [99,100]. This is a standard finding in motor learning, song acquisition in songbirds, and hippocampal cognitive mapping [101,102]. Thus, the Go/Go paradigm provides a robust platform for detecting these shifts and offers a rapid, reliable approach for studying auditory learning and its neural basis. A combination of behavioral paradigms and computational modeling offers a powerful strategy for advancing our understanding of evidence accumulation during sensory-driven learning and its underlying neural mechanisms [103,104].

**Figure 9 animals-15-03452-f009:**
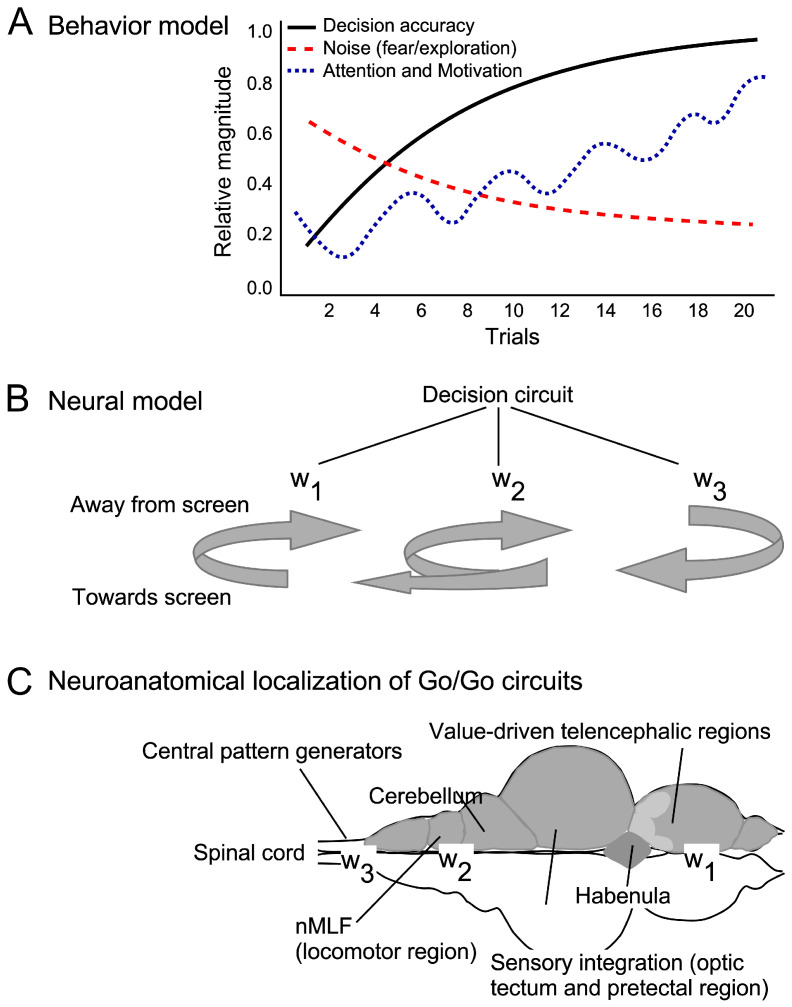
(**A**) Line plots for a conceptual representation of factors influencing decision-making and swim patterns [103]. Learning results in increase in attention and motivation that typically fluctuate in a periodical manner (dotted line). Behavioral noise decreases gradually (dashed line) as the decision-making processes are refined over successive runs (solid black line). (**B**) At the neural level, refinement of decision-making can be considered as changes in the synaptic weights (w1, w2 and w3) as the fish swims towards and away from the screen in response to the sound presented until a significant differentiation of the synaptic weight distribution between different brain regions is achieved. This distribution represents different states of the neural circuits related to motivation and movement. (**C**) A dorsal view of the zebrafish brain showing the location of different brain regions involved in the Go/Go response to different audiovisual stimuli.

### 4.5. Neural Circuits Underlying Motivated Swimming and Decision-Making

Motivated swimming and decision-making are likely regulated by an interplay between brain regions associated with fear, reward, and motor control (Figure 9C). The caudoventral portion of the medial pallium, the lateral and dorsal pallium, and striatal regions provide tonic inhibition to a diencephalic motor nucleus that projects directly to reticulospinal neurons [105]. Turning off this inhibition can trigger swimming. The ventral telencephalon (homologous to the mammalian striatum) plays a critical role in processing rewards and mediating goal-directed behaviors [106]. Dopaminergic projections from the posterior tuberculum, which resemble the mesolimbic reward pathway in mammals, are crucial for reinforcing behaviors associated with positive outcomes [107]. These centers have been shown in other species to receive direct auditory inputs.

The habenula and its connections with the interpeduncular nucleus regulate value-driven behaviors, including fear and aversive learning by modulating downstream serotonergic and dopaminergic activity contributing to the avoidance of the aversive stimulus (bullfrog video) in our study [108,109,110]. The balance between reward and fear-processing within these circuits may explain the initial variability in zebrafish swim patterns, as fish dynamically weigh competing motivational drives during early learning. These projections likely drive the adjustment in synaptic weights that motivate fish to swim towards the rewarding stimulus in response to the sound type (CF or FM) presented. A deeper understanding of these neural circuits can provide insights into the value- and attention-driven basis for overcoming behavioral noise.

The pretectal area, optic tectum, and cerebellum contribute to sensory integration, decision-making, and motor control (Figure 9C). Additionally, the cerebellum and torus semicircularis contribute to refining motor patterns and encoding auditory stimuli, respectively [10]. Additional brain centers for processing auditory stimuli have been detailed in studies in zebrafish larvae and other fish species [10,63]. The cerebellum’s role in motor learning is particularly relevant in our paradigm, as it allows zebrafish to adjust the timing of their swim trajectories in response to changing stimuli. The torus semicircularis has been shown to encode FM sounds, providing a sensory basis for discriminating between the two sound types used in this study [10,63]. The optic tectum, responsible for processing visual and auditory stimuli, likely integrates auditory cues and visual US. This integration informs the nucleus of the medial longitudinal fasciculus that in turn recruits reticulospinal neurons and biases central pattern generators within the spinal cord to trigger turning movements that facilitate goal-directed swimming [111,112].

### 4.6. Methodological Applications and Future Directions

The concept of a Go/Go assay can be applied to train almost any species both efficiently and effectively. Both the CS and the US can be adapted to meet the needs of the experimenter and species in question. For example, fruit flies can be trained to respond to different sounds using a brief exposure to an attractive vs. a repulsive odor as the US [113]. Similarly, rodents can be trained to respond to odors and other behaviorally meaningful stimuli within customized arenas [114]. Generally, applicable tracking programs, such as DeeplabCut [115] and specialized tracking programs for worms, flies, and mice can provide sensitive tracking to identify subtle but reliable changes in behavior, especially direction of locomotion. By building on dynamic models, future studies could further refine the analysis of zebrafish behavior in the Go/Go paradigm [116]. For example, incorporating time-varying covariates into predictive models could enhance our ability to dissect the contributions of different neural and behavioral states to learning. Such approaches could also be used to test the effects of pharmacological agents on specific phases of learning, from initial shifts in synaptic weights to long-term memory consolidation.

Our apparatus and software can be customized to advance clinically relevant research. A commercial development of the apparatus and assay proposed and used in this study can allow one to test multiple animals to facilitate rapid, high throughput screening of drugs designed to alleviate learning, memory, and attention deficits, and track development of auditory capacity of peripheral and central hearing. These drugs can be either injected or simply added to the tank water. The Go/Go learning assay can also provide a relatively inexpensive method to study how learning alters gene expression in the brain. For example, sequencing relevant brain regions involved in auditory learning can elucidate the role of synaptic proteins and microRNAs [117]. Single-cell RNA sequencing paired with associative conditioning can identify cell types present in value- vs. sound-driven brain regions. Similarly, neuronal populations that play a role in arousal, anxiety and decision-making can be identified at the genetic level [118]. Receptor function can be studied by injecting drugs to block receptors and directly observing their effects on sensory processing, motor activity, and learning behaviors within our learning paradigm. Furthermore, employing social reward, as used here, can be used to test social motivation and evaluate drug interventions for neurodevelopment disorders such as autism spectrum disorders since zebrafish and humans carry some of the same genes, e.g., *Shank3*, whose absence or mutation can lead to autism-like deficits.

For advancing our understanding of the neural underpinnings of behavior, two-photon imaging can be used to track neural activity triggered by sounds vs. attention after training GCaMP-labeled animals within the Go/Go paradigm on the same or next day [106,119]. Newly developed methodologies even allow optogenetic activation and visualization (e.g., using CaMPARI) of neurons active during different learning phases in behaving animals [120]. Finally, by using iontophoretic, optogenetic, and chemo-genetic methods to make localized ablations, future studies could explore the role of specific neurons and pathways [121]. For example, one could dissect the relative contributions of the auditory and lateral-line systems to complex sound discrimination, as might occur during shoaling behavior [122].

The setup and assay used here can be further improved by minimizing screen-side bias by adjusting lighting to eliminate reflections. Further refinements to reduce behavioral noise and identify a zone of best contrast prior to running the assay can allow additional manipulations within this zone to study complex behavioral phenomena, e.g., the effect of attention and aging on the sensory profile and movement characteristics. These efforts will enhance our understanding of zebrafish behavior and ecology and inform research on sensory processing and memory in other vertebrates, including humans.

## 5. Summary and Conclusions

In this study, we developed and tested a new paradigm that facilitates rapid learning in zebrafish. Our so-called Go/Go paradigm highlights the efficiency of naturalistic approaches to behavioral assays in combination with markerless tracking methods using machine learning approaches. We used the assay to show that at the behavioral level, adult zebrafish can perceive and discriminate between complex sounds. Which acoustic parameter or combination of parameters (such as, duration and number of individual sound segments or silence intervals, bandwidth, spectral constitution) they cue on for sound discrimination needs further investigation. We showed that by listening to the two sounds, fish can predict a social reward vs. an aversive stimulus and accordingly move either towards or away from the screen displaying these stimuli. Adult zebrafish learn to do this within 90 min, facilitating the combining of additional interventions and imaging of neural imaging, as well as high-throughput screening of drugs for memory, cognitive, and movement disorders. We also demonstrated overnight memory consolidation, suggesting that the rapid learning is retained. Incorporation of this methodology in other species together with genetic and neural manipulations has the potential to open new avenues of research in understanding attention and learning mechanisms in the brain.

## 6. Patents

A patent application for the development of the Go-To/Go-Away training paradigm and assay, as well as custom apparatus is pending. Assignee: Georgetown University.

## Figures and Tables

**Figure 1 animals-15-03452-f001:**
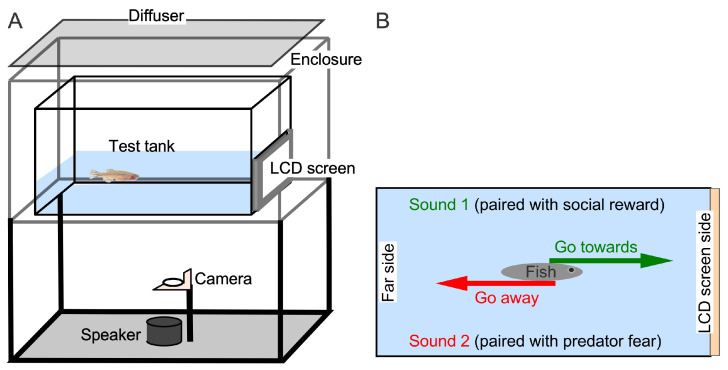
(**A**) Diagrammatic representation of the side view of the behavioral training setup with placement of the LCD screen on one side, and the webcam and Bluetooth speaker under the test tank. A translucent sheet covered the tank to diffuse ambient light and provide uniform illumination. (**B**) Top view of test tank to show the design of the approach/avoidance conditioning assay. One sound (CF) is associated with swimming toward a socially rewarding visual stimulus displayed on a screen, while the other sound (FM) is associated with withdrawal.

**Figure 2 animals-15-03452-f002:**
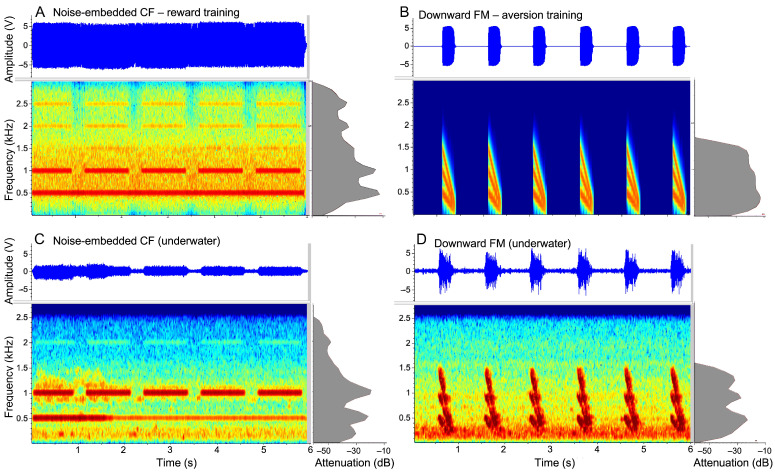
Amplitude envelopes (**top**) and spectrograms (**below**) of the two types of sound stimuli presented to the fish. The power spectrum of each sound is shown as grey shaded area in the right margin. (**A**) Train of digitally synthesized CF (NCF) tones with four harmonics embedded within narrowband noise. (**B**) Pulses of six downward frequency modulation (DFM) sweeps with equivalent energy in the three harmonics. (**C**,**D**) show the amplitude envelopes and spectrograms of the same two sounds as recorded with the hydrophone to show a ~14 dB loss of amplitude but not of the acoustic structure of the complex sounds.

**Figure 4 animals-15-03452-f004:**
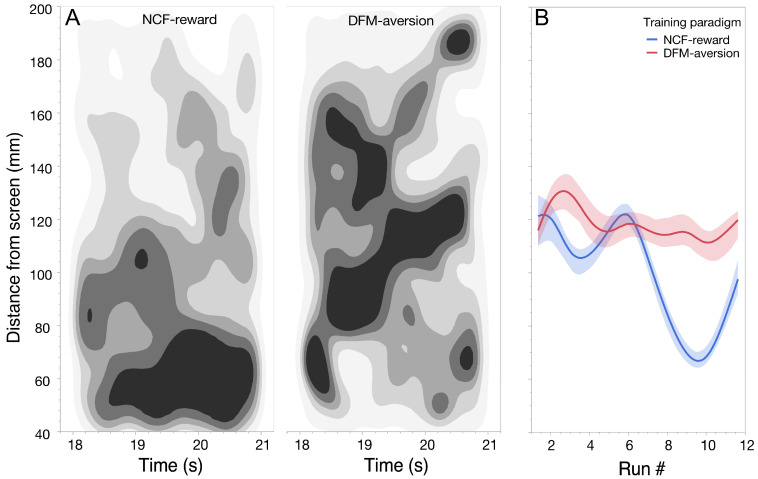
(**A**) Density plots of average locations across all training runs (1 through 12) for all fish (n = 21) during the 3 s gap interval between the end of the presentation of the two sounds (NCF and DFM) and playback of the video clip on the LCD screen. Fish locations at 40 mm from the screen and the far-end of the tank were excluded from the calculation to eliminate effects of screen-side bias and any edge-effects, respectively, associated with fish lingering along the tank walls. (**B**) Median spline fit of fish locations during the gap interval averaged across all 21 individuals showing divergence of fish position over time. Shaded zones indicate a 90% confidence interval.

**Figure 5 animals-15-03452-f005:**
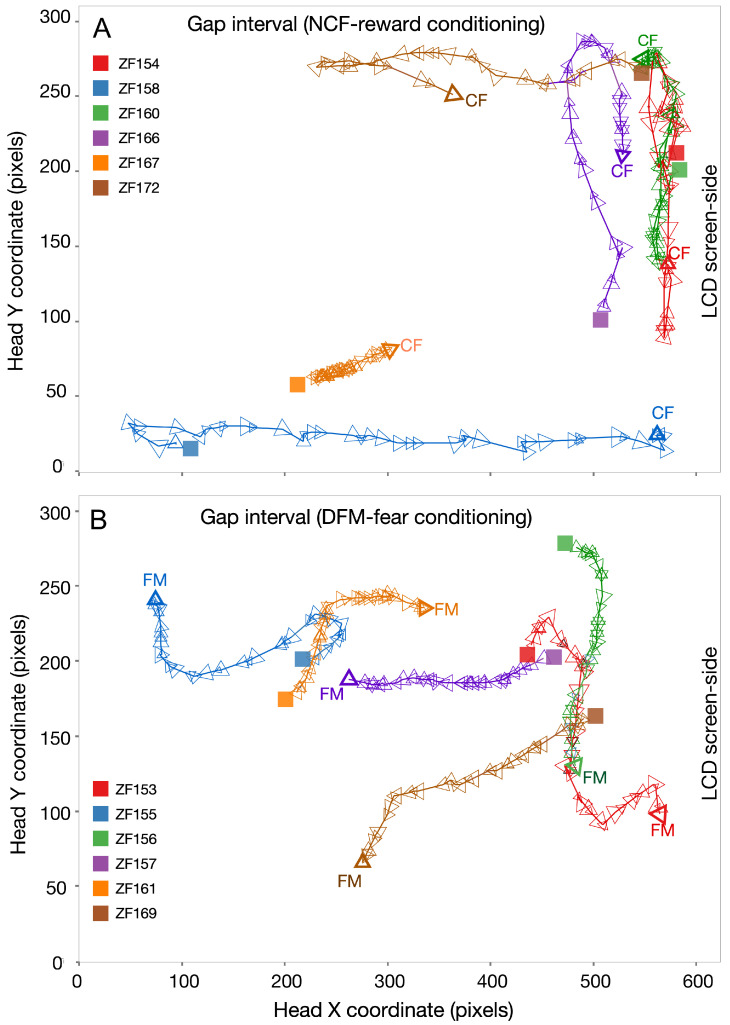
Timeline bubbleplots with trails to show the track of each animal’s color-coded swim trajectory during the 3 s gap interval. Tracks shown are superimposed from the first test run (run #13). (**A**) Tracks to the NCF*_rew_* in six animals and (**B**) to the DFM*_fear_* in a different set of six animals. Vertex of each triangular marker indicates direction of movement within a video frame, and triangle size is proportional to the relative value of acceleration at that location. Filled squares indicate the starting position and bold triangles, labeled as the “CF” or “FM”, indicate the terminal position at the end of the same-colored track. The vertices of triangles along the tracks indicate the direction of the tracked fish movement during each video frame. A larger space between triangles translates to a higher swim speed. The LCD screen was present on the right at X-value of ~600.

**Figure 6 animals-15-03452-f006:**
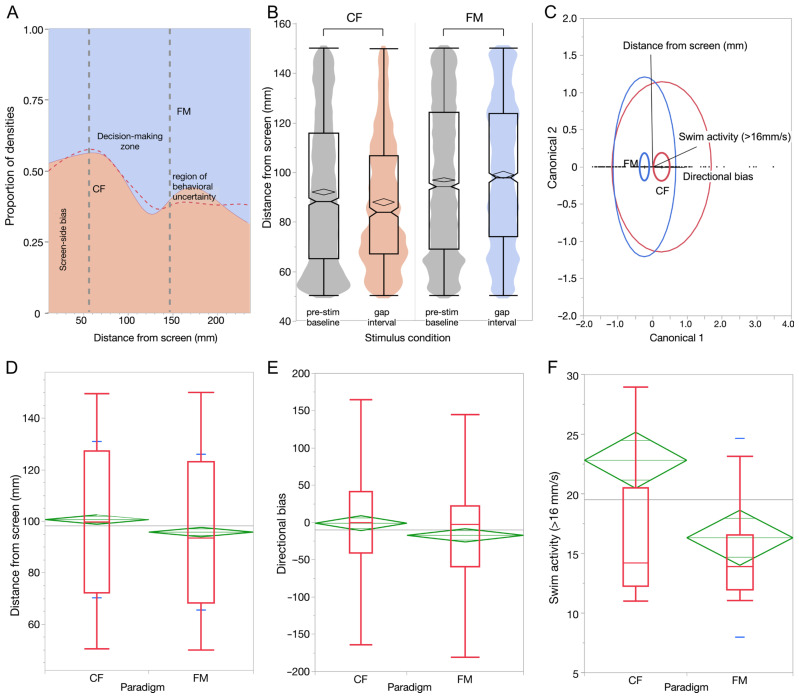
Behavioral parameters describing complex sound discrimination in zebrafish. (**A**) Proportion of density plot for test runs 13 to 18 to show the relative proportions of fish locations for the CF vs. the FM sound cues at different distances from the screen. The first gray line (~50 mm from screen) indicates a boundary of potential screen-side bias. The second vertical dashed line defines a decision zone (~50 to ~150 mm from the screen) that was used for all analyses. The dashed red line separates the two areas for a similar plot for data from runs 7 to 12. The resulting difference between the solid and dashed lines is labeled as a “region of behavioral uncertainty” during randomly alternating presentations of the two sound stimuli during the test runs. (**B**) Notched boxplots superimposed on violin plots to compare respective distributions of distances from the screen during the pre-stimulus baseline condition vs. the gap period after either the FM or CF presentation (runs 7 to 12). Notches correspond to the median values and confidence diamonds indicate the upper and lower 95% of the mean. Whiskers extend to the last point that is within 1.5 x inter-quartile range from the ends of the box. (**C**) Centroids and normal 50% contours in the multivariate space generated from DA to test for discrimination between CF*_rew_* (red oval) and FM*_fear_* (blue oval) test runs from the pooled, within-group covariance matrix captured by the three parameters (factor loadings indicated by labeled rays in the plot). (**D**–**F**) Box plots showing comparisons for FM and CF data for each of the three parameters used for DA. A negative value for FM for “directional bias” (calculated for the first 2 s of the gap interval), indicates that fish moved away from the screen. Green diamonds represent the mean and 95% confidence interval for each group for test runs. Horizontal red and blue lines indicate medians and standard deviations, respectively, and thin gray line is the grand mean.

**Figure 7 animals-15-03452-f007:**
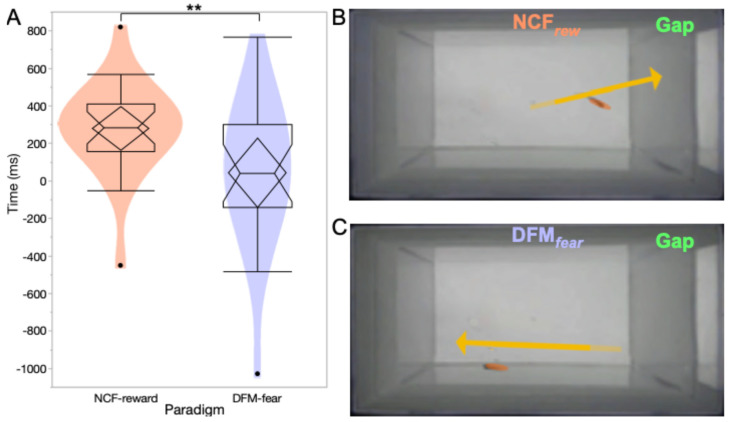
(**A**) Notched boxplots comparing NCF and DFM sound presentations for duration of longest trajectory for NCF*_rew_* vs. DFM*_fear_* for runs 10 to 18 (n = 16). Negative values represent movement away from the screen and positive values toward the screen during the gap interval. (**B**,**C**) Single frames from videos of fish trajectories in response to the two sound presentations. Test tank is imaged from below and screen is on the right side. Fish image is highlighted in orange oval and arrows represent beginning and end of trajectories (see Appendix A for videos). ** *p* < 0.01.

**Figure 8 animals-15-03452-f008:**
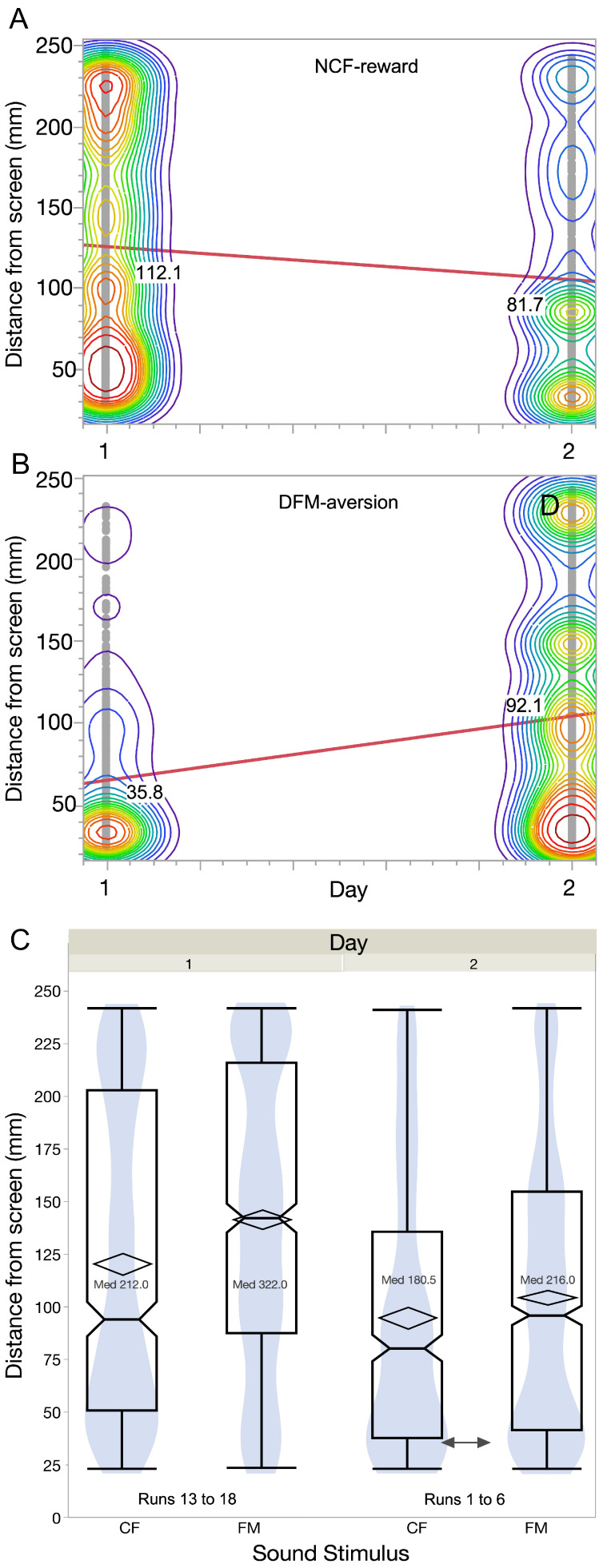
Bivariate smoothed density plots with contours at quantiles across two days for NCF*_rew_* (**A**) and DFM*_fear_* (**B**) conditioning to show overnight memory consolidation. The regression lines of fit and median values for fish location are indicated on the plots. (**C**) Notched boxplots superimposed on density plots for distance from screen in response to presentation of the NCF*_rew_* vs. the DFM*_fear_* sounds (left panel) on Day 1 of conditioning. Diamonds represent the mean and 95% confidence intervals. Similar plots (right panel) for Block 1 runs presented on Day 2 to test retention of conditioning. Note: Frame-level data are not independent samples and are shown only for descriptive visualization. All inferential statistics are performed on per-fish summary values.

**Table 1 animals-15-03452-t001:** Counts and average values of three behavioral parameters used in the DA.

Count	Paradigm	Distance from Screen (mm)	Swim Activity (>16 mm/s)	Directional Bias
102	CF	98.579	17.487	61.316
118	FM	100.758	13.500	−48.870
220	All	99.748	15.349	2.216

## Data Availability

The data acquired in this study can be made available by the corresponding author upon request.

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
