# Peer review of "Complex Sound Discrimination in Zebrafish: Auditory Learning Within a Novel “Go/Go” Decision-Making Paradigm"

_animals, 2025, doi:10.3390/ani15233452_

Round 1

Reviewer 1 Report (Previous Reviewer 1)

Comments and Suggestions for Authors

Line 16: "including mammalian, species" should be: including mammalian species, have an erroneous comma
Lines 137-138: Animal numbers are unclear. "We used a total of 55 fish (21 male; 13 female, 21 undetermined; age range: 3 to 12 months). Thirty-four fish were used for assay development." It is critical to clarify if the 34 fish used for development are a subset of the 55, or an additional group. Furthermore, the final test uses 21 fish. The justification for the sample size in the final test (n=21) and especially the memory consolidation test (n=7) should be explicitly stated, perhaps with a note on statistical power.
Line 138: The punctuation in the fish count is inconsistent. Should be (21 male, 13 female, 21 undetermined; age range: 3 to 12 months). And the 3-month fish should be juvenile, but the 12-month fish may be adult fish. The author ought to clarify the presence of any effects in the results.
Line 145: "with water to the test tank" should be "with water into the test tank"
Line 181: “Figures 1C and 1D show the spectrograms…” should be Figures 2C and 2D
Lines 225: The block design (e.g., six consecutive NCF-reward trials followed by six DFM-aversion trials) risks introducing order effects (e.g., fatigue, loss of motivation, contrast effects). The authors mention a "lingering or hysteresis effect" (Line 370), which acknowledges this possibility. The manuscript should discuss whether the order of these blocks was counterbalanced across fish and how potential carry-over effects were considered in the analysis.
Line 242: "played on an LCD screen" is less standard than "displayed on an LCD screen" for visual stimuli.
Line 368, L416: The term "confusion" is informal and subjective. Consider replacing it with a more neutral and descriptive term like "region of behavioral uncertainty"
Line 447: "Kolmogrov-Smirnov" is misspelled. Should be: Kolmogorov-Smirnov. The use of the Kolmogorov-Smirnov test for comparing distributions of directional bias is noted. However, this test is sensitive to sample size. To ensure the robustness of this finding, the authors should provide extra supporting statistics or effect sizes together with the p value, particularly due to the small sample size.

Author Response

Reviewer 2 Report (New Reviewer)

Comments and Suggestions for Authors

The manuscript aims to investigate zebrafish locomotor responses to two types of sounds and associated learning involving a specific visual cue, utilizing markerless tracking methods combined with machine learning approaches. To enhance clarity, the following specific points should be addressed:

The rationale behind the timing of stimulus presentation (was any delays between sound and visual cues?) and the selection of short intervals (e.g., CF-GAP-FM or FM-GAP-CF) is not sufficiently justified. Please clarify why these specific intervals were chosen and how they relate to the learning process. Additionally, specify how many events or repetitions occurred during each conditioning trial.

Were fish tested only once per block or multiple times? How many trials were conducted for each CS-US pair, and were individual fish tested repeatedly? Was the procedure different during “single block” training sessions for overnight memory consolidation?

Why was the frog image used as an unconditioned stimulus? Is there evidence that the fish recognize this model as a predator, especially if they have not previously encountered such images or cues? How consistent is this recognition, considering the frog model's inflated sacs and presentation in an air environment? Furthermore, it remains unclear whether the FM signals used in the study are comparable to those naturally produced by predators or represent biologically relevant predator cues.

Regarding the transgenic fish (Huc:GCaMP6): no data or results related to this genetic modification are presented. If transgenic fish were used, please specify their purpose and include relevant results.

The experimental setup involves artificially generated sounds and stimuli that do not fully represent natural environmental cues, limiting ecological relevance. Consider referring to these conditions as “mesocosm-like” or “semi-natural” rather than fully naturalistic (see Lines 12, 503, 594, 730).

The placement of the sound source under the bottom of the test apparatus and visual cues on the aquarium walls may influence the fish’s responses. Fish could detect hydrodynamic cues or associate the sound source’s position with danger, confounding the interpretation of sound-visual associations. Please clarify whether the design might affect the responses and how potential confounds were mitigated.

The mention of “error-correcting behavior” (Lines 615-616) lacks clarity. Specify the type of error e.g., approach versus avoidance and whether the goal was to approach or evade the visual cues. Clarify what behavior is considered an error in this context.

Line 136. What was the amount of feed provided per day or per treatment group, expressed as a percentage of body weight? Additionally, what were the average size and weight of the fish?

Line 137. What exactly is “habitat water”? Was conditioned tap water used? Please specify.

Lines 140-142. Were the same tanks used for both maintenance and transport? Clarify the procedures.

Line 211. Please clarify the abbreviations CS (Conditioned Stimulus) and US (Unconditioned Stimulus). Additionally, were the experimental blocks distinguished solely by the sequences CF-R (CF paired with reward) and FM-A (FM paired with aversion)?

Line 486. I assume that the aquatic environment, covering roughly 70% of Earth's surface, is not unique but differs from terrestrial or aerial environments.

Round 2

Reviewer 2 Report (New Reviewer)

Comments and Suggestions for Authors

Dear Authors,

Thank you for your efforts and for addressing my comments. The overall merit of your work is high and highly relevant. However, some of your responses indicate that the authors may have been confused by my simplified questions, and I feel it is necessary to clarify certain points.

First and foremost, a correct description of the experimental design should be understandable to a broad range of researchers. Therefore, in the introduction and M&M section, it would be advisable to explain the rationale behind choosing the experimental approach, specifically the principle of the Go-to/Go-away paradigm applied in this study. While I agree with the authors that a 6-second response window is sufficient, the primary aim appears to be the development of a method suitable for application in future pharmacological studies. Such applications require a strictly standardized approach with a clear and comprehensible protocol, enabling easy interpretation of results. Hence, it is essential to justify the choice of the decision-to-action interval within the block construction, as well as the selection of stimuli used.

It remains unclear what was used as a control in the study. How did you differentiate the fish’s reactions to predator images from potential responses to variations in screen brightness alone? Additionally Line 134-135, 227, 666, 1341, 1358, 1899, 1923. The term “devising a new naturalistic assay” may be inaccurate, as the experimental approach employed does not fully replicate a natural or ecologically valid context. Please revise accordingly. The study was conducted under traditional laboratory conditions, involving several features that deviate significantly from a natural environment, including manual manipulation of the fish, the use of filtered tap water, a transgenic model, the presentation of artificially synthesized stimuli (both sounds and visual cues), along with other unpredictable features. Please use more appropriate terminology to accurately characterize these stimuli.

Line 141-144. The manuscript lacks crucial detail regarding the model organism. It is unclear whether wild-type, a stable transgenic line, or adult fish with transiently incorporated Huc:GCaMP6 constructs were used. Clarification of the specific genetic background and methodology employed for GCaMP6 delivery is needed. While it is evident that your study aimed to investigate brain ion channel function during behavioral responses, the current manuscript does not explicitly address this topic. Therefore, for clarity, it is necessary to clearly state the exact fish type used. Additionally, the age of the fish used in the experiments was not reported and should be included.

The manuscript also does not specify the lighting conditions within the test apparatus or whether they differed from the maintenance tanks. Such differences could act as potential stressors. Furthermore, protocol details should be clarified by stating the duration of the fish’s acclimation period in the test apparatus before each trial.

Author Response

We thank you for taking the time to read our revisions and evaluate our manuscript. We are very appreciative of the attention to clarity and standardization, and constructive feedback on further improvements in the presentation of our findings. We are delighted by your notation that “the overall merit of our work is high, and it is highly relevant”. Building on your feedback that prompted some rethinking, we have improved the scientific accuracy and provided necessary details with better wording.

Comment 1: First and foremost, a correct description of the experimental design should be understandable to a broad range of researchers. Therefore, in the introduction and M&M section, it would be advisable to explain the rationale behind choosing the experimental approach, specifically the principle of the Go-to/Go-away paradigm applied in this study. While I agree with the authors that a 6-second response window is sufficient, the primary aim appears to be the development of a method suitable for application in future pharmacological studies. Such applications require a strictly standardized approach with a clear and comprehensible protocol, enabling easy interpretation of results. Hence, it is essential to justify the choice of the decision-to-action interval within the block construction, as well as the selection of stimuli used.

Response 1: Our purpose for this study is two-fold. First and foremost, as is clear from the Introduction, to use adult zebrafish as an established model to demonstrate that fish can discriminate between complex sounds. Given our development of a novel assay for this purpose, we delineate its development in the supplementary section and discuss it’s potential for being used for pharmacological and other learning-related studies. Establishing a strictly standardized approach or data analysis pipeline for future pharmacological studies is not a focus of this work because that will need to be dictated by the goals and authors of future studies. Publication of this manuscript and additional funding, however, may allow us to increase efficiency of the setup, such as testing three animals simultaneously, and make hardware improvements in the future. Regardless, as stated, the description of the experimental design should be correct and understandable to a broad range of researchers. To facilitate this, we now include Fig. 1B illustrating the principle of approach/avoidance in the “Go-to/Go-away conditioning assay. We also add the following justification under Method sections 2.3 and 2.5, respectively.

“Our behavioral paradigm uses a “Go-to/Go-away” associative learning framework in which the fish learns to differentially respond to two auditory stimuli by either approaching or withdrawing from a visual target screen. Together with markerless tracking this design enables measurement of continuous behavioral trajectories rather than binary choices, allowing one to characterize the moment-to-moment evolution of behavioral state dynamics during learning. Rather than forcing a correct/incorrect response at a discrete time point, the paradigm preserves the fish’s natural ability to swim freely, producing rich, time-resolved behavioral signals that allow quantification of learning in terms of trajectory shape, spatial proximity, and steering dynamics.

“Statistically, the experiment was structured as a within-subject repeated-measures design. Each fish experienced both stimuli across matched trial blocks, permitting paired comparison of behavioral state evolution rather than between-group contrasts. This approach increased statistical sensitivity and reduced inter-subject variability. Thus, the Go-to/Go-away paradigm provided the continuous kinematic structure necessary to evaluate the stability and learning-dependent reorganization of behavioral policy.”

The 3 s gap (response window) selection was explained in the previous revision. Per your suggestion, we have added the following as rationale for our experimental approach and justification of the 6 s CS (listening window). The modified text in the Introduction now reads as follows:

“Bullfrog vocalizations can last for nearly one second and are generally repeated five to seven times (Simmons 2004). They contain both noisy and frequency-modulated acoustic elements. We therefore hypothesized that adult zebrafish perceive and discriminate between complex sounds, such as frequency-modulated (FM) and constant-frequency (CF) sounds. Accordingly, we synthesized FM and CF sound stimuli that were well within the auditory spectrum of zebrafish hearing, as established in previous studies [62]. They were acoustically comparable and contained within the upper and lower bounds of frequency bandwidth and duration of bullfrog vocalizations. To test our hypothesis, we used both a social reward and an aversive unconditioned stimulus for associative conditioning.”

With respect to the Go-to/Go-away paradigm as utilized in our research, we added the following in the Methods section 2.2 (citations are included under References in the finalized version of the manuscript): 

“For testing auditory discrimination ability of zebrafish in our setup, we exploited the natural behavior of freely behaving animals. We reasoned that in their natural environment animals can move towards sources of stimuli that are attractive and away from those that are threatening. Therefore, in contrast to the Go/No-Go training paradigm where animal movement is inhibited (Popper et al. 2019; Mishkin and Pribram 1955), we let zebrafish swim freely when encountering either a rewarding or fearful stimulus (Fig.1B).”

We also added the following under Methods, 2.2:

“In summary, stimulus selection (NCF vs DFM) was constrained to the zebrafish hearing range and matched in energy; FM rates were designed to be consistent with amphibian vocalization envelopes, providing ecologically equivalent complexity while remaining synthetic to avoid familiarity confounds.”

To restate content previously included in lines 263 to 272, – as part of standardization of timing and order, we used six consecutive CS–US pairings per block, three blocks per session and one session per fish. To summarize the rationale succinctly, the Go-to/Go-away (Go/Go) paradigm is a classical conditioning design in which a conditioned auditory stimulus (CS) predicts either a socially rewarding or an aversive unconditioned visual stimulus (US) presented at the same screen location. During the 3 s gap after the 6 s CS, fish either approach or withdraw from the screen according to learned CS–US associations, enabling discrimination readouts with minimal handling and naturalistic swimming.

Comment 2: It remains unclear what was used as a control in the study. How did you differentiate the fish’s reactions to predator images from potential responses to variations in screen brightness alone?

Response 2: As implied previously, in our design, each animal acts as its own control with the sound (CF vs. FM) being the conditioned discriminant. Visual US stimuli successfully evoked approach and avoidance and this was their primary purpose. Any variations in screen brightness, even if present, are not particular relevant within the context of CS reinforcement if they are effective and used consistently. In our experimental design, the CS was always auditory; both the rewarding (shoal) and aversive (frog) US videos were presented on the same LCD screen, below the waterline, with comparable luminance settings and without adjusting brightness. We have now added (see Methods 2.3):

“Our primary readout is behavior during the 3 s gap after the CS and before any video appears; thus, the discrimination metric is independent of and exhibited in the absence of the visual display. Within-subject counterbalancing (block order randomized across fish) further controlled for any residual visual bias, such as subtle differences possibly in US brightness.”

We explicitly state:

“Screen luminance and placement were constant across US conditions and that analysis windows exclude the US epoch”.

Comment 3: Additionally Line 134-135, 227, 666, 1341, 1358, 1899, 1923. The term “devising a new naturalistic assay” may be inaccurate, as the experimental approach employed does not fully replicate a natural or ecologically valid context. Please revise accordingly. The study was conducted under traditional laboratory conditions, involving several features that deviate significantly from a natural environment, including manual manipulation of the fish, the use of filtered tap water, a transgenic model, the presentation of artificially synthesized stimuli (both sounds and visual cues), along with other unpredictable features. Please use more appropriate terminology to accurately characterize these stimuli.

Response 3: We agree with your enumeration of several assay features that deviate significantly from a natural environment. Our usage of “naturalistic”, however, implied its dictionary meaning and stated usage as in “The zoo strives to create naturalistic settings for the animals”. We used “naturalistic” to imply design elements within our assay that allow spontaneous, unrestrained swimming and ethologically relevant cues (social shoal video; complex FM/CF sounds) without head-fixation or forced-choice arms. This usage aligns with standard definitions (e.g., ‘naturalistic settings’ to approximate natural conditions) and with common practice in behavioral ecology. We avoid claiming a fully natural habitat (i.e., we do not say ‘natural’). In this sense, our usage was consistent with previous literature that use “naturalistic” to describe their assays (e.g., see Gerlai, 2010*).

In short, usage of naturalistic is a bit nuanced. To deemphasize its usage, we have eliminated the use of “naturalistic” in lines134-135 and 227. We also eliminated “mimic” and “naturalistic” in line 666 because these terms suggest that our model closely replicates real-world phenomena, whereas our intention was to indicate a partial resemblance. We have also eliminated the sentence at line 1341, modified the one on line 1358, and eliminated “naturalistic” on lines 1358 and 1899. We retained its usage at line 1923 where it refers to an approach that we strive for.

*Gerlai R. (2010) Zebrafish antipredatory responses: a future for translational research? Behav Brain Res. Mar 5;207(2):223–31.

Comment 4: Line 141-144. The manuscript lacks crucial detail regarding the model organism. It is unclear whether wild-type, a stable transgenic line, or adult fish with transiently incorporated Huc:GCaMP6 constructs were used. Clarification of the specific genetic background and methodology employed for GCaMP6 delivery is needed. While it is evident that your study aimed to investigate brain ion channel function during behavioral responses, the current manuscript does not explicitly address this topic. Therefore, for clarity, it is necessary to clearly state the exact fish type used. Additionally, the age of the fish used in the experiments was not reported and should be included.

Response 4: We now state that the fish used were a stable transgenic line (Huc:GCaMP6) bred from wild-type strain of zebrafish and obtained from the laboratory of Dr. Burgess at the National Institutes of Health, Bethesda, MD. This and other such lines are widely available for zebrafish research and are maintained via standard, established procedures in the zebrafish core facility. Fish were visually confirmed to have normal appearance and swimming patterns prior to testing. (see Methods, 2.1)

The age range of the fish that were used was included in lines 150-151 in response to a comment during the previous review.

Comment 5: The manuscript also does not specify the lighting conditions within the test apparatus or whether they differed from the maintenance tanks. Such differences could act as potential stressors. Furthermore, protocol details should be clarified by stating the duration of the fish’s acclimation period in the test apparatus before each trial.

Response 5: Lines 182-183 in the submitted version described the laboratory lighting conditions that were used during data collection. The light cycle in the core facility was also described in lines 141-142. We now state explicitly that the quality of the light used in the two locations was generally similar. Acclimation period in test tank was previously indicated but we have further elaborated on these details as follows (see Methods 2.1).

“Fish were acclimated in the laboratory environment for 15 to 30 minutes before they were gently transferred with a transparent, water-filled scoop to the test tank and kept in it for at least 5 minutes to allow acclimation and exploration of the tank environment prior to initiation of conditioning. The temperature and quality of lighting in the experimental room generally matched that in the core facility. “

Comment 6: Line 16: "including mammalian, species" should be: including mammalian species, have an erroneous comma.

Response 6: This was deleted after previous review and revision and not present in the submitted version; perhaps, its deletion was not clear in the tracked version. 

Comment 7: Lines 137-138: Animal numbers are unclear. "We used a total of 55 fish (21 male; 13 female, 21 undetermined; age range: 3 to 12 months). Thirty-four fish were used for assay development." It is critical to clarify if the 34 fish used for development are a subset of the 55, or an additional group.

Response 7: We previously corrected this to state, “Thirty-four of these were used for assay development (detailed in the Supplementary section).”.

Comment 8: Furthermore, the final test uses 21 fish. The justification for the sample size in the final test (n=21) and especially the memory consolidation test (n=7) should be explicitly stated, perhaps with a note on statistical power.

Response 8: Thank you for bringing this to our attention. The sample size of n = 21 for the main discrimination experiment was determined based on prior behavioral work in adult zebrafish for examining conditioned approach/avoidance responses under similar effect sizes (Singh et al., 2022; Aoki et al., 2015). In our dataset, the primary directional bias metric produced moderate to large effect sizes (Cohen’s d = 0.47–0.89 depending on comparison), indicating that the observed effects are robust at this sample scale. Our statistical analyses were conducted on a within-subject design, which increases power by reducing between-individual variance. We now add the following statements under Methods 2.5 as justification:

“… For the final discrimination test performed with n = 21 fish, the sample size was determined based on prior behavioral work in adult zebrafish examining conditioned responses (Singh et al. 2022; Valente et al. 2012)) and deemed typical for within-subject behavioral learning paradigms in adult zebrafish.

“Overnight consolidation was quantified as changes in state-space dispersion in Block 1 on each day The memory consolidation subset (n = 7) consisted of fish drawn from the same trained cohort and served as a retention confirmation assay rather than an independent inferential test, consistent with prior zebrafish conditioning studies (e.g.,(Aoki et al. 2015)).”

The memory consolidation test was an add-on to the main study. Here, we used Block 1 on each day to provide a standardized “initial” behavioral state from which overnight consolidation can be quantified as changes in state-space dispersion. The within-fish comparison, however, did not apply here given the small sample size and an unbalanced design since conditioning was only compared for the first block on each day to test for consolidation. Because of randomized presentation on each day, the first block condition could be either CF-reward or FM-fear in the same animal. As a result, we do not have enough data to perform to test significance for within-fish comparisons as we did for the main study. Contiguous frames/samples do not satisfy the condition of independence and should not be pooled across fish to avoid oversampling. Accordingly, we cannot compute statistical power. We have removed p values indicating significance but retain figures as descriptive visualizations suggesting a trend towards memory consolidation.

To strengthen our conclusion of memory consolidation, we now quantify a state-space framework that is both established and analytically valid for the behavioral question being addressed. Our behavioral paradigm developed here directly measures state transition structure and thus is aligned with current frameworks emphasizing policy stabilization and reduction in exploratory variance as signatures of memory consolidation that we already touched upon in our discussion section (section 4.4). We specifically note (Methods section 2.5):

“… We quantified state-space dispersion within the first six trials of each day (Block 1), which represents the nascent behavioral state. For each fish, we computed the variance of proximity (distance to the goal) and steering (yaw), aligned to the direction of the conditioned response, and used the trace of the covariance matrix as a scalar measure of the total variance in exploratory or conditioned behavior. A high trace indicates a high total spread across all variables, while a low trace indicates a low spread.”

Under Results (section 3.4), we added:

“For an across-animal statistical test for memory consolidation, we used a trial-block state analysis approach where each fish contributed a high-dimensional behavioral trajectory. We observed a consistent reduction in dispersion between Day 1 and Day 2 (Wilcoxon signed-rank, p = 0.25; rank-biserial effect size (rrb) = 0.80; n = 4), reflecting a large effect size and a strong and consistent directional shift in behavioral variability across fish on day 2 (see Fig. S8). In such within-subject designs, statistical power is governed by consistency across individuals, not sample size in the classical between-group sense. This analysis indicated that behavior became more stable and less exploratory after the overnight period, even when mean approach distances remained comparable. Such a reduction in trial-to-trial variability is a well-established signature of memory consolidation and policy stabilization in both motor and associative learning frameworks (e.g., (Dudai 2012)). Here learning is inferred from shape and stability of behavioral manifolds rather than binary performance metrics. This statistical result reflects strong within-individual consistency.”

These results conform nicely with our discussion on a state-based model for auditory learning, so we also added the following insect. 4.4:

“Our data indicated a qualitative shift from a more widely distributed and variable state-space occupation on the first day of training to a more compressed and directionally focused state on the second day. This reflects reduced exploratory variance and a movement toward a more stable and efficient approach policy. When responses were collapsed to per-fish median values, across-fish Wilcoxon tests were not significant (p > 0.25), indicating that individual-level variability is high when temporal sequence information is removed, consistent with an early learning stage in which cue-response associations are emerging but not yet stabilized. Even though paired-sample medians did not differ significantly across days, a reduction in state-space dispersion is consistent with partial consolidation of the learned mapping between auditory cue and directional motor response. This is important, because consolidation often reduces variance before changing means (Garst-Orozco et al. 2014; Smith 2015). This is a standard finding in motor learning, song acquisition in songbirds, and hippocampal cognitive mapping (Rusu and Pennartz 2020; Brudner et al. 2023).”

This manuscript is a resubmission of an earlier submission. The following is a list of the peer review reports and author responses from that submission.

Round 1

Reviewer 1 Report

Comments and Suggestions for Authors

The manuscript describes a novel and interesting behavioral paradigm (Go/Go) and reports preliminary evidence that zebrafish can distinguish between two complex sounds and associate them with visual stimuli of different combinations. 

Line 2 The term 'Go/Go' itself may not be intuitive enough and can easily cause confusion. The author needs to provide a clearer explanation of its meaning (i.e. requiring an active response in opposite directions to both stimuli), or consider using another more appropriate term (such as "Approach/Avvoid" or "Go Way/Go Away"), although the latter may not be as concise as "Go/Go".

Line 181: Is the underwater microphone a hydrophone? Please provide the frequency response, sensitivity of the hydrophone, and instructions on how to record sound in the tank, for example the . Also, please provide the sound pressure level (SPL) and spectrum.

Line 186, should "ABCD" instead of "left" "top ",  and i was not understanding why the AB waveforms were different from the C D at the same time scales, if C D were the recording of A B,.  in Fig 2,  and airborne sound transmission may not replicate natural soundscapes.

L198, what is the ecology role of Bullfrog video. whether the sound is produced by the bullforg? there is a huge visual difference between rewarding US (zebrafish group) and aversive US (bullfrog). Fish may react and differentiate based on the visual features of the video itself (movement patterns, shapes, colors), while sound is only an accompanying, non essential clue. Experimental design requires strong evidence to prove that fish are indeed distinguishing sound itself, rather than differences in visual stimuli or fixed combinations of sound and specific visual stimuli.

L211 what means is CS, it seems to be the first time it has appeared,should provide the complete meaning. 

Author Response

Reviewer 1:

Comment 1: “The manuscript describes a novel and interesting behavioral paradigm (Go/Go) and reports preliminary evidence that zebrafish can distinguish between two complex sounds and associate them with visual stimuli of different combinations.”

Line 2 The term 'Go/Go' itself may not be intuitive enough and can easily cause confusion. The author needs to provide a clearer explanation of its meaning (i.e. requiring an active response in opposite directions to both stimuli), or consider using another more appropriate term (such as "Approach/Avoid" or "Go Way/Go Away"), although the latter may not be as concise as "Go/Go".

Response 1: We thank the reviewer for the positive remark and time spent to think of an improved name of the paradigm. The Go/Go was meant to be a play on the standard “Go/NoGo” training paradigm. We chose this term because it is concise and more likely to catch-on. Per your suggestion, we now refer to it as Go-to/Go-away decision-making paradigm abbreviated as “Go/Go” paradigm. We have deleted the Latin name for zebrafish in the title to keep it within two lines.

We hope our evidence of sound discrimination is more than “preliminary” and open to confirmation by others. Our manuscript included details of the development of the novel assay, and the term “preliminary results” was used within that context. This resulted in some confusion regarding the strength of findings in the final analysis, so to avoid any confusion in the reader’s mind, we have moved these data and their explanation to Supplementary information under “Methodology development and analysis”.

Comment 2: Line 181: “Is the underwater microphone a hydrophone? Please provide the frequency response, sensitivity of the hydrophone, and instructions on how to record sound in the tank, for example the . Also, please provide the sound pressure level (SPL) and spectrum.”

Response 2: Yes, the underwater microphone we refer to is a commercially available hydrophone (DE-PRO, DolphinEar Global) that has an omnidirectional configuration with a wide frequency-response of 1 Hz to 24 kHz. We now consistently use the term hydrophone and include information on its response characteristics together with instructions on how to record sounds and measure the sound pressure level in the tank in the Supplementary section. The manufacturer of the DE-PRO does not provide a calibration sheet, therefore, at this time, we are unable to provide the exact SPL. Both sounds tested were equalized in amplitude using the RMS values and spanned a similar frequency range, so should not have a direct bearing on sound discrimination data. We have included the power spectrum of both sounds alongside their spectrograms in Fig. 2, as suggested.

Comment 3: Line 186, should "ABCD" instead of "left" "top ", and i was not understanding why the AB waveforms were different from the C D at the same time scales, if C D were the recording of A B,. in Fig 2, and airborne sound transmission may not replicate natural soundscapes.

Response 3: We have modified labels in the figure and improved its explanation in the figure legend. It now states:

Figure 2. Amplitude envelopes (top) and spectrograms (below) of the two types of sound stimuli presented to the fish. The power spectrum of each sound is shown in the right margin. A. Train of digitally synthesized CF (NCF) tones with four harmonics embedded within narrowband noise-. B. Pulses of six downward frequency modulation (DFM) sweeps with equivalent energy in the three harmonics. C and D show the amplitude envelopes and spectrograms of the same two sounds as recorded with the hydrophone to show an ~ 14 dB loss of amplitude but not of the acoustic structure of the complex sounds.”

Comment 4: L198, “what is the ecology role of Bullfrog video. whether the sound is produced by the bullforg? there is a huge visual difference between rewarding US (zebrafish group) and aversive US (bullfrog). Fish may react and differentiate based on the visual features of the video itself (movement patterns, shapes, colors), while sound is only an accompanying, nonessential clue. Experimental design requires strong evidence to prove that fish are indeed distinguishing sound itself, rather than differences in visual stimuli or fixed combinations of sound and specific visual stimuli.”

Response 4: We took a naturalistic (ecological and ethological) approach towards developing our conditioning paradigm. As was indicated in figure 3, there is no sound associated with the presentation of the image of the frog. The sounds that are being tested are presented prior to the presentation of the frog video and end before the frog video appears on the screen. All testing of sound discrimination is based on fish behavior in a 3 sec. gap interval when no sound or video is present. The unconditioned stimuli merely need to be effective in producing a measurable response. A separate question currently being investigated relates to the most effective feature of the US for inducing fear. This appears to be a frog’s face compared to a visual of inflating and deflating vocal sacs (Patel et. al., in preparation).

Comment 5: L211 what means is CS, it seems to be the first time it has appeared should provide the complete meaning.

Response 5: We now provide, as we should have, the complete meaning for first time use of CS.

Reviewer 2 Report

Comments and Suggestions for Authors

My first review was not succecfully submitted, and now I'm busy, so I'll be brief.

Below are some saved text from the first review:

This work demonstrate experimentally that zebrafish has the ability to discriminate between complex sounds (constant frequency and frequency-modulated sound types ) via audiovisual associative conditioning. It may be a useful method also for future studies of invertebrate species or  other vertebrate species.

The study is innovative and seems well conducted

Weakness: I think Fig 4 can be better presented and it also includes some confusing or erroneous text.

Strength: There is a considerable lack of knowledge about acoustic communication in fishes, and this study demonstrates that hearing is of importance in learning and behavioural decisions of zebra fish.

The study methods seems relevant for the future study of Join, leave and stay decision (JLS) in fish species.  

That is discussed in a series of paper – see below. The experiment provides strong evidence for the hypothesis that hearing and lateral line perception is crucial in JLS decision, since it showed that a complex sound can stimulate join decision.

References:

Larsson Schooling Fish from a New, Multimodal Sensory Perspective, Animals 2024

A reference, the book of Max Bennet "A Brief History of Intelligence" 2024 might well be added since it provide a good background about conditioned stimuli and the evolution of brains in vertebrates, mammals.

Author Response

Reviewer 2:

Comment 1: “This work demonstrate experimentally that zebrafish has the ability to discriminate between complex sounds (constant frequency and frequency-modulated sound types) via audiovisual associative conditioning. It may be a useful method also for future studies of invertebrate species or other vertebrate species. The study is innovative and seems well conducted.”

Response 1: We appreciate the positive remarks regarding the study and its significance. Also, per your in-line comments, we now use shortforms NCFrew and DFMfear for the two sounds throughout the manuscript and made additional minor improvements elsewhere, as suggested.

Comment 2: “Weakness: I think Fig 4 can be better presented and it also includes some confusing or erroneous text.”

Response 2: We apologize for the unclear presentation of figure 4 and its legend. Its lack of clarity was also noted by other reviewers. We have improved its quality and present only results related to sound discrimination. Previously, our manuscript included results obtained during the development of the methodology, which we thought were useful to detail. This led us to pack too much information in a condensed figure. We have moved parts A and B of this figure together with original figures 5 and 6 and related text to the supplementary section to improve readability.

Comment 3: “Strength: There is a considerable lack of knowledge about acoustic communication in fishes, and this study demonstrates that hearing is of importance in learning and behavioural decisions of zebrafish.”

Response 3: We agree; thank you.

Comment 4: “The study methods seems relevant for the future study of Join, leave and stay decision (JLS) in fish species. That is discussed in a series of paper – see below. The experiment provides strong evidence for the hypothesis that hearing and lateral line perception is crucial in JLS decision, since it showed that a complex sound can stimulate join decision.”

“References: Larsson Schooling Fish from a New, Multimodal Sensory Perspective, Animals 2024

A reference, the book of Max Bennet "A Brief History of Intelligence" 2024 might well be added since it provide a good background about conditioned stimuli and the evolution of brains in vertebrates, mammals.”

We have now explored these interesting references and cite Larson in the Methodological applications section of the Discussion. Our results do not demonstrate a direct connection between sounds and JLS decisions but make it more feasible to study using our assay, e.g., digital dissection of sounds may identify acoustic components associated with shoaling and similarly in predator sounds. Max Bennet’s book is certainly a stimulating read though too general a source to cite here.

Reviewer 3 Report

Comments and Suggestions for Authors

Overall the topic addressed by this paper is very interesting and could have some strong impact, however, the authors need to put a lot more work into this study, in the analysis of the data and in the writing of this paper.

Major problems include a number of overclaims, a low N value for some figures and inadequate statistical tests in some cases. The authors have not made enough efforts in their writing, it is difficult to follow their train of thoughts, there are typos everywhere, a lot of mis-referencing (e.g. wrong figure panel numbers). Some figures are difficult to see, some others have poor X or Y labelling.

In the following, I give more detailed feedback where I separate major and minor problems.

Major Comments:

  • Most of your abstract is about the experimental method. An abstract is not meant to describe a method but summarize the motivation, the results and their impact. You must rewrite the whole abstract with this in mind.
  • About the motivation of your abstract: complex sounds (e.g. white noise and frequency sweeps) have already been studies in zebrafish (Poulsen et al. Current Biology 2021) and their respective neural networks have been described. Results show that the networks are different and therefore, these complex sounds are processed differently in zebrafish. The behavioural responses/changes are not shown though. You discuss this latter in your introduction, however, these results mean that you should revise some of your claims and hypothesis accordingly: lines 11-12, 20-21, 57-58, 61-64, 784-786.
  • Line 177 and 179, there is no Figure 1B, 1C or 1D, please fix the numbering. I am guessing you mixed up Figure 1 and 2. Then Figure 2 legend is also a mess, e.g. line 188 “Top panel shows”? Which top panel? There are 2 top panels. Use your letters maybe?
  • Lines 299-301: “In the beginning, fish tend to explore their environment more widely compared towards the end of a session where they tended to stay towards the center.” How do you support this claim? Is this the result of just looking at Figure 4 A and B? This needs to be quantified and you need to show a significant difference. You should quantify these for all you fish.
  • Same thing for figure 4C and D. You need to quantify those for all the fish recorded to be able to claim what you are claiming about these.
  • I don’t think an N of 6 is sufficient for Figure 4. Strangely your method mentions higher numbers. I would think that an N of at least 10 starts to be reasonable. Fish are relatively easy to maintain, breed and handle. Is there any specific reason why you used only 6?
  • They are many typos through out the text that shows poor text revision from the authors. I mentioned a couple. But please fix them all.
  • In Figure 5 you show the quantification of the distance from screen across the run number for 1 fish from each group only. And from that you conclude: “These plots show that place preference exhibited in response to a sound is not a reflexive 350 response but gradually learned so that during successive gap intervals, the fish positions 351 itself at an appropriate location prior to the onset of video playback for the US. In short, 352 fish predict the US from the sound and learn to swim either towards or away from the 353 screen over successive trials.” An n of 1 is not sufficient to support this claim. And Figure 5A by itself is not sufficient either. You should show in that figure 5B (or maybe in another one) the traces from all the fish and its average across runs.
  • I have similar concerns for Figure 6: n= 4 is too low, why this number? And you draw conclusions from 6B from 1 fish only from each group. You need to show all the fish traces in Figure 6B or in another panel in a different way.
  • “The data show that, on average, fish tend to spend significantly (P <0.05, n=4) more time near the screen”. I don’t understand where you got this p value from? You are showing us density plots in 6A, without actual numbers. How did you calculate this p value? You might want to add a panel with the data you do the statistical analysis on.
  • I don’t understand why you handpicked test trials 13 to 18 and trials 7 to 12 for figure 7A. Again, the display of the data seems very arbitrary and doesn’t represent the whole experiment. Maybe if you picked other trials, we wouldn’t see that effect at all. Because of that, you cannot make overall claims on your dataset.
  • Figure 7B: Using Willcoxon Two-Sample Rank Sums test (typo in Willcoxon) is not appropriate for your data. You should do a Wilcoxon Rank-Sum Test. Please explain your choice.

Minor comments:

  • Lines 54-55: “we have relatively little information and appreciation of the underwater acoustic scene”. Aren’t hydrophones meant to be used and precisely quantify these?
  • Line 84: “how 83 their survival may be threatened by a global warming of ocean and freshwater biomes” I don’t understand why the study of sound will help us towards this. I suggest you remove this.
  • Lines 95-96 are unclear and a backet is missing. Please revise.
  • Line99, rather than “chemical”, I would say “olfactory” (to stay consistent and describe senses rather than elements).
  • Line 112, References 58 and 59 are not doing 2P calcium imaging. It’s one photon in both papers. Please revise.
  • Line 117 “signals, They”. Capital T after a coma? Please revise. This is an obvious typo and should have been picked up by all authors. Please revise all your manuscript and correct all typos.
  • Line 119, add a coma after “work”. Please revise your punctuation through all your manuscript.
  • 14/10 light:dark is repeated twice: lines 126 and 131.
  • Please add the acronyms of the types of sound on their appropriate panels in Figure 2 for clarity.
  • Figure 3 is very blurry, your should improve its resolution.
  • Choose better colors for Figure 4, 2 fish are in green to me and cannot be distinguished.
  • Give the n number in Figure 4.
  • Figure 4 mentions Bubbleplots. I googled what it is supposed to look like, and it is not what you are showing. Use the correct term for your plots.
  • Units for X and Y axis in Figure 4 are missing.
  • Figure 4A in line should be Figure 5A? Please fix all other figure references. There are many mistakes.
  • Panels in Figure 5 and 6 are squeezed. Please fix this. Also the X axis labels are not consistently written. “Distance from” on the top right seems misplaced.
  • “As part of the exploratory analysis, we first generated proportion of densities 417 plots to estimate the relative density of fish locations for CF vs. FM sound cues during the 418 gap interval.” Where is this data? Please reference it at the end of your sentence.
  • Figure 7. Remove the “A” at the beginning of the figure caption, looks like a typo.
  • Figure 7A panel, the y axis label “Proportion” is a poor labelling. Proportion of what?
Comments on the Quality of English Language

The authors have not made enough efforts in their writing, it is difficult to follow their train of thoughts in many places, there are typos everywhere (most of them very obvious), a lot of mis-referencing (e.g. wrong figure panel numbers). I find some sentence difficult to understand too. The whole manuscript would benefit a lot from a careful writing revision.

Author Response

Reviewer 3:

Comment 1: “Overall the topic addressed by this paper is very interesting and could have some strong impact, however, the authors need to put a lot more work into this study, in the analysis of the data and in the writing of this paper”.

Response 1: We thank you for expressing your interest in our methodology and findings, and for your time and effort to provide constructive feedback to improve this manuscript. This has been extremely useful. A lot of work went into the development of the new paradigm for using it to obtain the results on auditory discrimination. We therefore wanted to share some of this effort. We have now put in significant effort in reorganizing the manuscript to improve presentation of the main results. We also added a new figure with a density plot representing data from all animals and re-wrote substantial portions, including the abstract.

Comment 2: “Major problems include a number of overclaims, a low N value for some figures and inadequate statistical tests in some cases. The authors have not made enough efforts in their writing, it is difficult to follow their train of thoughts, there are typos everywhere, a lot of mis-referencing (e.g. wrong figure panel numbers). Some figures are difficult to see, some others have poor X or Y labelling.”

Response 2: Thank you for pointing these problematic issues. We previously included preliminary results obtained during the development of the methodology, which we thought were useful to detail. This appears to have resulted in some confusion and perception of a low N and overclaims. These data were only meant to serve as guides for further development and modifications of our paradigm, and to allow authors a deeper insight into the assay in case others attempt to use or customize the novel assay. We apologize for not making this clear enough. We now focus solely on the results obtained via the final three-block design. To enhance readability, figures 4A, 4B, 5 and 6 and related text have been moved to the Supplementary section. This together with substantial re-writing of relevant sections and improving figure quality in both the main and supplementary sections has allowed us to improve the train of thought. Specific changes are noted below.

Major Comments:

Comment 3: “Most of your abstract is about the experimental method. An abstract is not meant to describe a method but summarize the motivation, the results and their impact. You must rewrite the whole abstract with this in mind.”

Response 3: Consistent with changes in the text of the manuscript and your suggestions, we have now revised/re-written the abstract for it to reflect the focus of the results and not include details about methodology development. It now reads as follows:

Abstract: Previous anatomic and physiologic studies of the peripheral and central auditory system, with rare exceptions, have relied on the use of tonal stimuli. Here, we test the hypothesis that zebrafish, Danio rerio, can detect and discriminate between two 6 s long complex sounds — a sequence of five multi-harmonic, noise-embedded constant frequency (NCF) tone pips and a chirp sequence of six rapid downward frequency modulated (DFM) sweeps. To test our hypothesis, we develop an associative conditioning assay, requiring prediction of an unconditioned stimulus (US). A video clip of a shoal of free-swimming zebrafish presented on an LCD screen serves as a desirable or rewarding US and a bullfrog with inflating and deflating vocal sacs serves as an undesirable or aversive US. Within our novel “Go-to/Go-away” assay, sound discrimination is critical for deciding to go/swim towards the desirable US and away from the undesirable US within a short time window preceding each US. Our results were visualized via markerless tracking of fish locations following twelve training runs. On average, fish move closer to the LCD screen in response to the sound paired to the rewarding CS and farther away from the screen in response to the sound paired with the aversive US. Fish also exhibit opposite-swim trajectories in response to the two sounds. These differences are retained on the second day of testing, suggesting overnight memory consolidation. We conclude that zebrafish can both perceive and rapidly learn to discriminate between complex sounds and that our novel assay can be implemented for high throughput screening of drugs targeted for alleviating memory and other neurodegenerative disorders.

Comment 4: “About the motivation of your abstract: complex sounds (e.g. white noise and frequency sweeps) have already been studies in zebrafish (Poulsen et al. Current Biology 2021) and their respective neural networks have been described. Results show that the networks are different and therefore, these complex sounds are processed differently in zebrafish. The behavioural responses/changes are not shown though. You discuss this latter in your introduction, however, these results mean that you should revise some of your claims and hypothesis accordingly: lines 11-12, 20-21, 57-58, 61-64, 784-786.”

Response 4: We acknowledge and respect the work that has been done previously on sound reception in fish species. This has allowed the field to move forward. Our research builds upon and supports the previous findings, and we do not intend to ignore or minimize their importance. We also have a responsibility to identify what is new in this study. We address each concern as detailed below:

Lines 11-12: “We experimentally demonstrate for the first time via audiovisual associative conditioning, the ability of adult fish to discriminate between complex sounds.”

Lines: 20-21: This is a specific hypothesis statement, so it is unclear how this should be modified. We are happy to consider any suggestions.

Lines: 57-58: We believe a lot more work can be done on sound perception in fish given the vast number of species living in different habitats. Our statement here was meant to stress the importance of conducting more research and not meant to ignore the work that has already been done in any way. We have added additional references here, that were included elsewhere, to avoid any misunderstanding.

Lines: 61-64: We have modified this text and instead state:  

“Fortunately, recent reports on auditory responses to pure- and spectrally-complex tones, white noise and courtship sounds, and on the organization of the auditory system are beginning to improve our understanding of sound perception in fishes [2,9–12]. “

Lines: 784-786: We now state: “We used the assay to show that at the behavioral level, adult zebrafish can perceive and discriminate between complex sounds.”

We have deleted the next sentence to avoid any misinterpretation.

Comment 5: “Line 177 and 179, there is no Figure 1B, 1C or 1D, please fix the numbering. I am guessing you mixed up Figure 1 and 2. Then Figure 2 legend is also a mess, e.g. line 188 “Top panel shows”? Which top panel? There are 2 top panels. Use your letters maybe?”

Response 5: We apologize for overlooking these problems after updating a couple of figures. We have made the necessary corrections.

Comment 6: “Lines 299-301: “In the beginning, fish tend to explore their environment more widely compared towards the end of a session where they tended to stay towards the center.” How do you support this claim? Is this the result of just looking at Figure 4 A and B? This needs to be quantified and you need to show a significant difference. You should quantify these for all you fish.”

Response 6: This was a general, unimportant statement that had no bearing on the results. This statement and the related portion of Figure 4 has been deleted (see below).

Comment 7: “Same thing for figure 4C and D. You need to quantify those for all the fish recorded to be able to claim what you are claiming about these. I don’t think an N of 6 is sufficient for Figure 4. Strangely your method mentions higher numbers. I would think that an N of at least 10 starts to be reasonable. Fish are relatively easy to maintain, breed and handle. Is there any specific reason why you used only 6?”

Response 7: Data were shown for 12 fish (six each for testing each sound, depending on which sound happened to be delivered first in the test trial (run # 13). Tracks shown are for illustrative purposes to give the reader a sense of the within and across group variation in swim patterns (raw data) during the gap interval. 

Comment 8: “They are many typos through out the text that shows poor text revision from the authors. I mentioned a couple. But please fix them all.”

Response 8: We have conducted a careful revision after re-writing and eliminating portions of the text and figures (see below) to make the manuscript streamlined and easier to follow and read. We are grateful for your comments that helped us to do so.

Comment 9: “In Figure 5 you show the quantification of the distance from screen across the run number for 1 fish from each group only. And from that you conclude: “These plots show that place preference exhibited in response to a sound is not a reflexive 350 response but gradually learned so that during successive gap intervals, the fish positions 351 itself at an appropriate location prior to the onset of video playback for the US. In short, 352 fish predict the US from the sound and learn to swim either towards or away from the 353 screen over successive trials.” An n of 1 is not sufficient to support this claim. And Figure 5A by itself is not sufficient either. You should show in that figure 5B (or maybe in another one) the traces from all the fish and its average across runs.

Response 9: The spline plots in original figure 5B were tracks of individual fish conditioned with either a CF-reward or an FM-aversion sound. As noted below for Figure 6, this was an illustration from a methodology development perspective. These data allowed us to gain insights into fish behavior, especially by visualizing fish locations in the jittered scatter plot. This information can be lost by averaging all or multiple fish. This figure was not meant to be the basis of making final conclusions from this study. In fact, single-stimulus (either FM or CF) conditioning sessions were superseded by the more challenging dual-stimulus training paradigm in the final version of this study. We indicated this in a couple of places in the manuscript and in ARRIVE guidelines. We have now eliminated any room for misunderstanding by moving this information and figure to the supplementary section.

Comment 10:”I have similar concerns for Figure 6: n= 4 is too low, why this number? And you draw conclusions from 6B from 1 fish only from each group. You need to show all the fish traces in Figure 6B or in another panel in a different way.”

Response 10: The original Figure 6B showed data from one fish from a successful training session to see how an individual behaves as part of methodology development. This information is sometimes lost if behavior from all animals is averaged. We have moved this figure to the Supplementary section (also see explanation below) and now show the average for all fish in the new Fig. 4B.

Comment 11: “The data show that, on average, fish tend to spend significantly (P <0.05, n=4) more time near the screen”. I don’t understand where you got this p value from? You are showing us density plots in 6A, without actual numbers. How did you calculate this p value? You might want to add a panel with the data you do the statistical analysis on.

Response 11: Thank you for expressing your concerns above related to data shown in figures 5 and 6 that could also appear in the mind of other readers. These figures and their explanation were designed to show sample raw (preliminary) data to explain the developmental steps of the new paradigm that we designed for this study. The significance test level for these data here was inappropriate and left over from a related poster figure. We have moved this figure and related text with some clarification to the Supplementary section under the subheading “Methodology development and analysis”. We have added the following text under section 2.3:

“Details of the development of this methodology and intermediate results are described in Appendix B of the Supplementary section. In brief, assay development started with a single sound stimulus block consisting of eight repetitions of the same CS-US pair. In separate animals, either NCF-aversion and DFM-reward or NCF-reward and DFM-aversion pairing was used. Exploratory free-swimming patterns were tracked during initial and final baseline conditions to be able to gauge the effect of conditioning (see Figs. S2A and S2B). Success with single CS-US pairing estimated from density plots for distance from screen (described in section B2 of the Supplementary section and illustrated in Fig. S3) was followed by a two-block design consisting of eight repetitions of NCF-reward (Block 1) and DFM-aversion pairs (Block 2) within a single conditioning session in the same animal. We first consecutively presented the same CS-US pair six times (Block 1) before switching to six repetitions of the alternate CS-US pair in Block 2. Thus, each training session included six consecutive presentations of either the CF or FM audio/visual pair (Block 1) followed by another six consecutive presentations of the alternate pair (Block 2). The order of presentation of the CF or FM audio/visual pairs was randomly selected across fish. These experiments were used to study block interaction effects on conditioning efficacy (see Figs. S4 and S5).

In the finalized design, a third block consisting of three NCF-reward (abbreviated as NCFrew) were randomly alternated with three DFM-aversion/fear (abbreviated as DFMfear) runs. Tracking data from these last six runs (#13 to 18) were used to test sound discrimination in 21 adult zebrafish. Thus, each animal was exposed to a total of eighteen runs per session. …”

Comment 12: “I don’t understand why you handpicked test trials 13 to 18 and trials 7 to 12 for figure 7A. Again, the display of the data seems very arbitrary and doesn’t represent the whole experiment. Maybe if you picked other trials, we wouldn’t see that effect at all. Because of that, you cannot make overall claims on your dataset.”

Response 12: Trials or “runs” 1 to 12 are the training runs in Blocks 1 and 2, whereas Block 3 (runs 13 to 18, consisting of random presentation of the NCF and DFM sounds) was used to test sound discrimination. This was indicated in the original version of the manuscript (e.g. see lines 216 to 218 and line 440). Interweaving methodology development description and data within the main body of the manuscript apparently contributed to some confusion. We have clarified the procedures as below:

“… Details of the development of this methodology and intermediate results are described in Appendix B of the Supplementary section. In brief, assay development started with a single sound stimulus block consisting of twelve repetitions of the same CS-US pair. In separate animals, either NCF-aversion and DFM-reward or NCF-reward and DFM-aversion pairing was used. Success with single CS-US pairing was followed by a two-block design consisting of eight repetitions of NCF-reward (Block 1) and DFM-aversion (Block 2) pairs within a single conditioning session in the same animal. We first consecutively presented the same CS-US pair six times (Block 1) before switching to six repetitions of the alternate CS-US pair in Block 2. Thus, each training session included six consecutive presentations of either the CF or FM audio/visual pair (Block 1) followed by another six consecutive presentations of the alternate pair (Block 2). The order of presentation of the CF or FM audio/visual pairs was randomly selected across fish. These data were used to study block interaction effects on conditioning efficacy (see Figs. S4 and S5’’).

In the finalized design, a third block consisting of three NCF-reward (abbreviated as NCFrew) were randomly alternated with three DFM-aversion/fear (abbreviated as DFMfear) runs. Tracking data from these last six runs (#13 to 18) were used to test sound discrimination in 21 adult zebrafish. … “

Response 12 (cont.): Figure 7A represents exploratory analysis (this was indicated on line 417) and includes a plot from conditioning runs 7 to 12 (dashed line) to compare effect of using random presentations during test runs with conditioning runs. We have further clarified this as follows:

“… As part of the exploratory analysis, we first generated proportion of densities plots separately for conditioning (Block 2) and test runs (Block 3) to estimate the relative density of fish locations for NCFrew vs. DFMfear sound cues during the gap interval (Fig. 6A). This provided us with a region of interest within the tank length where the fish appeared to be differentially active in response to the two sound types. …”

Comment 13: “Figure 7B: Using Willcoxon Two-Sample Rank Sums test (typo in Willcoxon) is not appropriate for your data. You should do a Wilcoxon Rank-Sum Test. Please explain your choice.”

Response 13: We have corrected the typo here and at other locations. A rank-sum test was indeed performed and is the general name for the more specific Willcoxon Two-Sample Rank Sums test, as named in JMPpro since two independent samples are involved.

Minor comments:

  • Lines 54-55: “we have relatively little information and appreciation of the underwater acoustic scene”. Aren’t hydrophones meant to be used and precisely quantify these?

Yes, they are. However, much more information needs to be obtained under different habitat conditions. We have modified this sentence along with the first two paragraphs of the Introduction to say:

“Hydrophones can be used to record underwater sounds, but intermittent noise and multipath propagation add to the difficulty of localizing sounds and obtaining spectrally clean recordings.”

  • Line 84: “how 83 their survival may be threatened by a global warming of ocean and freshwater biomes” I don’t understand why the study of sound will help us towards this. I suggest you remove this.

This part of the sentence has been removed.

  • Lines 95-96 are unclear and a backet is missing. Please revise.

Corrected. The sentence now reads: “Details about the natural history of zebrafish, including their environmental and predator-prey interactions have not been fully explored (however, see [56]).”

  • Line99, rather than “chemical”, I would say “olfactory” (to stay consistent and describe senses rather than elements).

Corrected to “chemosensory” and added appropriate references related to gustation.

  • Line 112, References 58 and 59 are not doing 2P calcium imaging. It’s one photon in both papers. Please revise.

Thank you for this information. We have corrected this.

  • Line 117 “signals, They”. Capital T after a coma? Please revise. This is an obvious typo and should have been picked up by all authors. Please revise all your manuscript and correct all typos.

“… auditory signals” was meant to be followed by a period, not a comma. We have tried our best to catch any others and hope anything still missed will be corrected by copy editor. We appreciate your attention to detail.

  • Line 119, add a coma after “work”. Please revise your punctuation through all your manuscript.

A comma after “work” did not seem appropriate; we have clarified the sentence. It now reads:

“An equivalent body of work in adult zebrafish, as was recently conducted in larvae, can provide useful insights …”

  • 14/10 light:dark is repeated twice: lines 126 and 131.

Information has been consolidated, and second occurrence has been deleted.

  • Please add the acronyms of the types of sound on their appropriate panels in Figure 2 for clarity.

Thank you for the suggestion. We have added these.

  • Figure 3 is very blurry, your should improve its resolution.

Screen shots of videos are generally not as sharp as a photograph because not all fish are in the focal plane at any one time. Other than that, the figure should be fine.

  • Choose better colors for Figure 4, 2 fish are in green to me and cannot be distinguished.

We worked extensively on this figure (Fig. 5 in revised version), overriding the JMP default color panel, which mismatched triangle and track colors, and improved contrast. We only kept the relevant gap interval plots to be able to increase size and clarity and moved the top panels showing baseline activity to the supplementary section.

  • Give the n number in Figure 4.

The animal # is included in the legend with some explanation.

  • Figure 4 mentions Bubbleplots. I googled what it is supposed to look like, and it is not what you are showing. Use the correct term for your plots.

These are in fact labeled as bubbleplots in JMPpro though the bubble size was greatly reduced to focus on tracks. We now label figure as “Timeline bubbleplots …” since they show animal movement over time.

  • Units for X and Y axis in Figure 4 are missing.

Units are in pixels. These have been added.

  • Figure 4A in line should be Figure 5A? Please fix all other figure references. There are many mistakes.

These have been corrected, and all references have been carefully checked.

  • Panels in Figure 5 and 6 are squeezed. Please fix this. Also the X axis labels are not consistently written. “Distance from” on the top right seems misplaced.

Corrected. This apparently happened because we missed to copy/paste the updated jpg versions in the MDPI template. Thank you for noticing. Font sizes now match, and letter labels are included for easy reference. These two figures are now in the supplementary section as Fig. S3 and S6.

  • As part of the exploratory analysis, we first generated proportion of densities 417 plots to estimate the relative density of fish locations for CF vs. FM sound cues during the 418 gap interval.” Where is this data? Please reference it at the end of your sentence.

Corrected. This referred to Figure 7A.

  • Figure 7. Remove the “A” at the beginning of the figure caption, looks like a typo.

Removed. Thank you.

  • Figure 7A panel, the y axis label “Proportion” is a poor labelling. Proportion of what?

Corrected: “Proportion of densties”.